# DDN: Dual-domain Dynamic Normalization for Non-stationary Time Series Forecasting

**Tao Dai**[1,*] **Beiliang Wu**[1,*] **Peiyuan Liu**[2,†] **Naiqi Li**[2,†] **Xue Yuerong**[2], **Shu-Tao Xia**[2], **Zexuan Zhu**[1]

[1]College of Computer Science and Software Engineering, Shenzhen University, China
[2]Tsinghua Shenzhen International Graduate School, Tsinghua University, China
{daitao.edu, peiyuanliu.edu, linaiqi.thu}@gmail.com; xiast@sz.tsinghua.edu.cn

## Abstract

Deep neural networks (DNNs) have recently achieved remarkable advancements in time series forecasting (TSF) due to their powerful ability of sequence dependence modeling. To date, existing DNN-based TSF methods still suffer from unreliable predictions for real-world data due to its non-stationarity characteristics, *i.e.,* data distribution varies quickly over time. To mitigate this issue, several normalization methods (e.g., SAN) have recently been specifically designed by normalization in a fixed period/window in the time domain. However, these methods still struggle to capture distribution variations, due to the complex time patterns of time series in the time domain. Based on the fact that wavelet transform can decompose time series into a linear combination of different frequencies, which exhibits distribution variations with time-varying periods, we propose a novel Dual-domain Dynamic Normalization (DDN) to dynamically capture distribution variations in both time and frequency domains. Specifically, our DDN tries to eliminate the non-stationarity of time series via both frequency and time domain normalization in a sliding window way. Besides, our DDN can serve as a plug-in-play module, and thus can be easily incorporated into other forecasting models. Extensive experiments on public benchmark datasets under different forecasting models demonstrate the superiority of our DDN over other normalization methods. Code is available at https://github.com/Hank0626/DDN.

## 1 Introduction

Deep neural networks (DNNs) with powerful dependency modeling capability have recently been widely used in time series forecasting (TSF) applications, including weather prediction [1], energy consumption estimation [2], and traffic flow forecasting [3]. Despite the great advancements of DNN-based TSF methods [4, 5, 6, 7], they still suffer from unreliable predictions for real-world data due to its non-stationary nature of real-world time series, *i.e.,* data distribution within the series varies quickly over time (*a.k.a*, distribution drift [8, 9, 10]). Such non-stationary challenge limits the real applications of DNN-based TSF methods.

To mitigate the problem of distribution drift, the classic reversible normalization [11] has recently been proposed with a two-stage pipeline of normalization and de-normalization. The former stage of normalization eliminates non-stationary factors for converting a non-stationary sequence into a stationary sequence, which has to acquire the mean and standard deviation of the sequence before. The latter stage of de-normalization reconstructs non-stationary information from the distribution prediction model or directly reuses the mean and standard deviation acquired in normalization.

---

*Equal contribution
†Correspondence to: Peiyuan Liu and Naiqi Li

38th Conference on Neural Information Processing Systems (NeurIPS 2024).

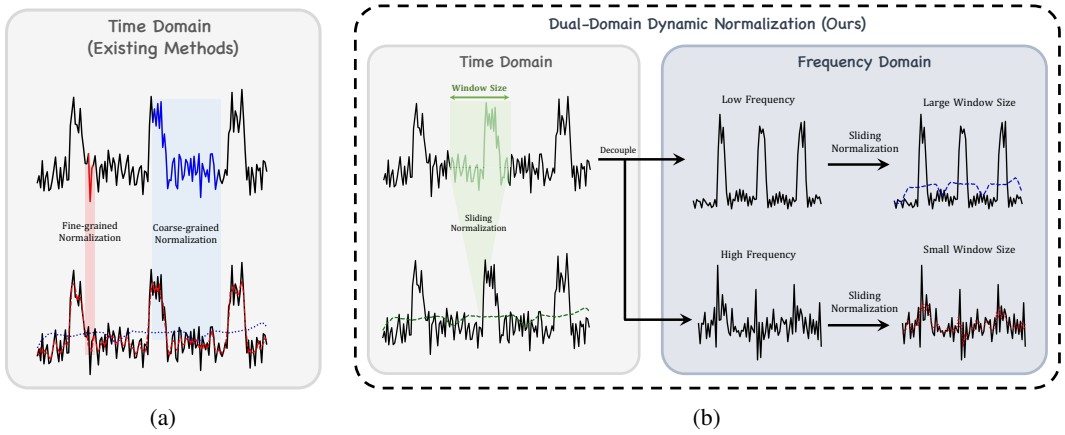

Figure 1: (a) Existing methods with a fixed period/window normalization struggle to capture distribution variations. (b) Our method dynamically captures distribution variations in both time and frequency domains.

Later, several advanced variants of reversible normalization [12, 13, 14] have achieved impressive performance by further alleviating the non-stationary property of real-world time series.

Despite the great success of normalization methods, existing methods are limited in capturing distribution variations by performing normalization with a fixed period/window. As shown in Figure 1a, either existing coarse-grained (e.g., RevIN [11]) or fine-grained normalization (e.g., SAN [14]) in single time domain tends to produce sub-optimal performance. On the other hand, it is known that wavelet transform can decompose time series into a time-dependent sum of frequency components, which exhibits distribution variations with time-varying periods (see Figure 1b). Thus, making full use of such frequency information is helpful to capture distribution variations with time-varying periods and intensities. These observations motivate us to develop a more powerful normalization strategy to dynamically capture distribution variations.

In this paper, we propose a novel **Dual-domain Dynamic Normalization (DDN)** framework to dynamically capture distribution variations in both times and frequency domains in a sliding window way. Specifically, our DDN decomposes the original time series into different frequency components, including low-frequency and high-frequency components, based on Discrete Wavelet Transform (DWT) [15, 16]. Followed by performing sliding normalization in an individual frequency component with proper window size (see Figure 1b), which is helpful to capture distribution variations with time-varying periods and intensities. Besides, time domain normalization is developed to compute local sliding statistics [17], including sliding mean and sliding standard deviation. Unlike the previous works that process a coarse-grained level, our DDN leverages fine-grained a more informative sliding window to calculate distribution characteristics for every time step.

Our main contributions can be summarized as: **(i)** We propose a novel Dual-domain Dynamic Normalization (DDN) to dynamically capture distribution variations in both time and frequency domains with sliding statistics. Compared with previous works, our DDN is capable of dynamically reflecting the rapid variations to time series. **(ii)** Our DDN aims to eliminate non-stationary factors with frequency domain normalization and time domain normalization. Benefiting from the complementary properties of the time and frequency domain information, it allows our DDN to further clarify non-stationary factors and reconstruct non-stationary information. **(iii)** Extensive experiments demonstrate the effectiveness of our DDN, by achieving significant performance improvements across various baseline models on seven real-world datasets.

## 2 Related Works

### 2.1 Deep Models for Time Series Forecasting

Reviewing the development of time series forecasting based on deep models, early methods [18, 19, 20] often integrated cross-dimension information in embedding module, then modeling cross-time

information. In contrast, recent Sota methods indicate that two modeling ways can be better: CI (Channel Independent) and CD (Channel Dependent). The primary distinction between the two approaches lies in the former focusing only on cross-time features but the latter incorporating cross-dimension features. Theoretically, the latter can leverage more information and achieve higher prediction accuracy [21, 22, 23]. In practice, for relatively short input series, CD methods [24, 25] achieve comparable or even better performance than the CI methods. However, for longer input sequences, the situation is often the opposite[26, 4, 27]. In recent research, this difference can be attributed to the CD having higher capacity but often lacking robustness in predicting distributional drift than CI [28, 29], while longer series typically experience more severe distributional drift. The superior performance of CI highlights the importance of handling distribution drift, and it is a valuable direction in the current research on time series forecasting.

## 2.2 Stationary for Time Series Forecasting

RevIN [11] was the first work to apply reversible normalization for time series forecasting, which assumes that history and future sequences share the same distribution. It counts distribution statistics of historical sequence for both normalization and de-normalization. Due to its simplicity and impressive effectiveness, it has been widely used in recent works [30, 31]. However, RevIN overlooks the distributional differences between historical and future sequences. Building upon RevIN, Dish-TS [12] proposes different distribution characteristics for historical and future sequences, using a distribution forecasting model to predict mean and standard deviation. Concurrently, NST [13] employs a module to provide more consistent distribution with future distribution, which can refer to appendix B. Furthermore, SAN [14] notes that existing distribution assumptions may not adapt to the scenario that time series points rapidly change over time [32, 33] and proposes a more fine-grained method, which supposes the distribution characteristics of time points is different between slices but same within a slice. Nevertheless, SAN still stops at the slice level, rather than the time series point level. Meanwhile, existing works lack consideration of the discrepancies between low and high frequencies, leading to insufficient consideration of non-stationary information.

# 3 Methodology

In the realm of multivariate time series forecasting, we consider a historical sequence $\boldsymbol{X} \in \mathbb{R}^{M \times L}$ and aim to predict the corresponding future sequence $\boldsymbol{Y} \in \mathbb{R}^{M \times T}$, $M$ is the number of channels. DDN is a model-agnostic plugin designed to align the distribution characteristics of $\boldsymbol{X}$ and accurately estimate the distribution of $\boldsymbol{Y}$. In this section, we will comprehensively outline the pipeline of the entire framework and elaborate on how to remove and reconstruct non-stationary factors of time series. To enhance clarity and facilitate understanding of subsequent chapters, the key notations used in this paper are summarized in Table 1, and the framework can be referred to in Figure 2.

| Notation | Description |
|---|---|
| $L, T$ | The time steps of the historical/future sequences |
| $\boldsymbol{x}^i, \boldsymbol{y}^i$ | The $i$-th historical or future series |
| $\bar{\boldsymbol{x}}^i_*, \bar{\boldsymbol{y}}^i_*$ | The $\boldsymbol{x}^i$ after normalization and it predicted series |
| $\boldsymbol{\mu}^i, \boldsymbol{\sigma}^i$ | The $i$-th mean or standard deviation series of $\boldsymbol{x}^i$ |
| $\boldsymbol{\mu}^i_y, \boldsymbol{\sigma}^i_y$ | The $i$-th mean or standard deviation series of $\boldsymbol{y}^i$ |
| $\boldsymbol{\mu}^i_*, \boldsymbol{\sigma}^i_*$ | The distribution forecasting of $\boldsymbol{\mu}^i$ or $\boldsymbol{\sigma}^i$ |

Table 1: Summary of key mathematical notations

## 3.1 Overall Framework

As depicted in Figure 2, we first eliminate non-stationary factors via both the Frequency Domain Normalization (**FDN**) and the Time Domain Normalization (**TDN**). These processes output two stationary sequences and two sets of distribution characteristics. Two stationary sequences weighted to a sequence and input to the time series Forecasting Model (**FM**) for future sequence forecasting, while two sets of non-stationary factors input to Distribution Prediction Model (**DPM**) and predict future non-stationary factors. Finally, these factors are weighted together and incorporated with forecasting

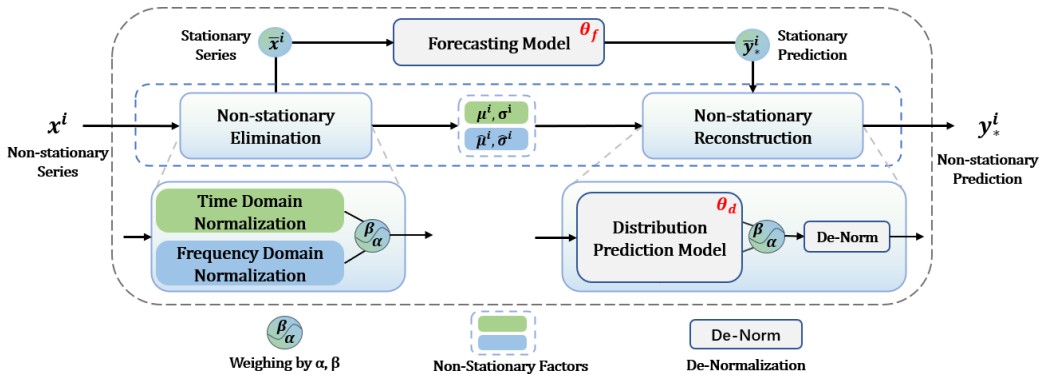

Figure 2: The comprehensive time series forecasting framework comprises a time series forecasting model and an auxiliary module designed for handling non-stationary factors. This auxiliary module consists of two sub-modules: one for eliminating non-stationary factors and another for reconstructing them. The non-stationary factor elimination sub-module includes Time Domain Normalization and Frequency Domain Normalization, while the non-stationary factor reconstruction sub-module incorporates a distribution prediction module.

output to reconstruct non-stationary factors by de-normalization. Here, $\theta_d$ and $\theta_f$ correspond to the parameters of DPM and FM, and the training strategy can be seen in the section 3.4.

## 3.2 Non-stationarity Elimination

For each series $x^i$, we perform a sliding window along the temporal dimension to acquire distribution characteristics, then replicate padding that will align the length of sliding statistics to the original series. Finally, sliding mean $\mu^i$ and sliding standard deviation $\sigma^i$ represent to the distribution characteristics of $x^i$. This process can be described as follows:

$$
\mu_j^i = \frac{1}{2k+1} \sum_{-k}^{k} x_{j+t}^i, \quad \left(\sigma_j^i\right)^2 = \frac{1}{2k+1} \sum_{-k}^{k} \left(x_{j+t}^i - \mu_j^i\right)^2,
$$

$$
\mu^i = \text{Pad}(\{\mu_{k+1}^i, \cdots, \mu_{L-k}^i\}), \quad \sigma^i = \text{Pad}(\{\sigma_{k+1}^i, \cdots, \sigma_{L-k}^i\}). \tag{1}
$$

Here $2k+1$ is the size of the sliding window, and stride is 1. After that, the size of sliding statistics is $L - 2k$. Where $\mu_j{}^i$ and $\sigma_j{}^i$ represent the mean value and standard deviation value of the $j^{th}$ time point respectively, where $j \in \{k+1, \cdots, L-k\}$. To make sure each time point possesses corresponding sliding statistics. We copy the sliding statistics closest in time by $\text{Pad}(\cdot)$ operation, the obtaining $\mu_i$ and $\sigma_i$ are used to achieve the transformation from non-stationary sequences to stationary sequences. The process is as follows:

$$
\bar{x}^i = \frac{1}{\sigma^i + \epsilon} \odot \left(x^i - \mu^i\right), \quad \epsilon > 0. \tag{2}
$$

Here, $\bar{x}^i$ is the stationary series, $\epsilon$ is a positive number to prevent the denominator from zero, and $\odot$ denotes the element-wise product. By this sliding normalization, annotated as $\text{SlidingNorm}(\cdot)$, we can acquire the non-stationary factors of each time point and convert non-stationary sequences to stationary sequences.

**Frequency Domain Normalization.** In this branch, to exhaustively unveil non-stationary factors and eliminate them accurately. Discrete Wavelet Transform (DWT) is conducted on $x_i$ to separate the low-frequency component $x_l^i$ and high-frequency component $x_h^i$. Subsequently, we acquire and eliminate their non-stationary factors. The process is as follows:

$$
x_l^i, x_h^i = \text{DWT}_{\phi_{l,h}}(x^i),
$$

$$
\bar{x}_l^i, \mu_l^i, \sigma_l^i = \text{SlidingNorm}(x_l^i), \quad \bar{x}_h^i, \mu_h^i, \sigma_h^i = \text{SlidingNorm}(x_h^i), \tag{3}
$$

Here, $\phi_{l,h}$ is a pair of learnable wavelet bases. $\bar{x}_l^i$, $\mu_l^i$, and $\sigma_l^i$ represent the stationary sequence, sliding mean, and sliding standard deviation of the low-frequency component. While $\bar{x}_h^i$, $\mu_h^i$, and $\sigma_h^i$

denote those of the high-frequency component. In practice, different types of DWT have different padding lengths and lead to different output lengths. To ensure a consistent and clear output length, Inverse Discrete Wavelet Transform (IDWT) performs to restore a definite size. The process is as follows:

$$\hat{\boldsymbol{x}}^i = \text{IDWT}_{\phi_{l,h}}(\bar{\boldsymbol{x}}^i_l, \bar{\boldsymbol{x}}^i_h), \quad \hat{\boldsymbol{\mu}}^i = \text{IDWT}_{\phi_{l,h}}(\boldsymbol{\mu}^i_l, \boldsymbol{\mu}^i_h), \quad \hat{\boldsymbol{\sigma}}^i = \text{IDWT}_{\phi_{l,h}}(\boldsymbol{\sigma}^i_l, \boldsymbol{\sigma}^i_h). \quad (4)$$

Where $\hat{\boldsymbol{x}}^i$, $\hat{\boldsymbol{\mu}}^i$, and $\hat{\boldsymbol{\sigma}}^i$ encompass the stationary sequences, sliding means, and sliding standard deviations of different frequency components. Through these operations, the output stationary sequence and distribution statistics maintain consistency with the dimensions of the input non-stationary sequence.

**Time Domain Normalization.** We conduct the same manners in the time domain without frequency decomposition. The process can be formulated as follows:

$$\bar{\boldsymbol{x}}^i, \boldsymbol{\mu}^i, \boldsymbol{\sigma}^i = \text{SlidingNorm}(\boldsymbol{x}^i), \quad (5)$$

The wavelet transform in FDN typically involves padding, which can potentially distort the statistical distribution information of the decomposed sequences. To address this, we implement sliding normalization directly on the original sequence. Consequently, the resulting distribution information is utilized for predicting future distributions, while the output stationary sequence is weighted with the stationary sequence derived from FDN.

**Stationary Sequences Weighting.** Two stationary sequences from FDN and TDN will be weighted to a stationary output, which can be expressed as follows:

$$\bar{\boldsymbol{x}}^i = \bar{\boldsymbol{x}}^i \cdot \beta + \hat{\boldsymbol{x}}^i \cdot \alpha. \quad (6)$$

Here, $\alpha$ is a trainable parameter and $\beta = 1 - \alpha$. The weighted $\bar{\boldsymbol{x}}^i$ serves as the final stationary sequence, which is then inputted into FM for stable forecasting.

### 3.3 Non-stationarity Reconstruction

We acquire two sets of sliding statistics reflecting distribution variations after FDN and TDN. Later, we refer to the structure of existing distribution prediction works [34, 12] to predict future distribution. Initially, we calculate the mean value of each sliding statistic to compute the statistical differences. Subsequently, the difference and original series are inputted for future difference prediction. Ultimately, these predicted differences added to the mean value as future sliding statistics.

**Frequency Domain Prediction.** As shown in Figure 3, for the distribution statistics of FDN, we predict future statistics by distribution prediction model and formulate as:

$$\begin{aligned}
\hat{\boldsymbol{\sigma}}^i_\Delta &= \text{SP}\left(\hat{\boldsymbol{\sigma}}^i - \sigma^i_f, \boldsymbol{x}^i\right), \quad \hat{\boldsymbol{\sigma}}^i_* = \hat{\boldsymbol{\sigma}}^i_\Delta + \sigma^i_f, \\
\hat{\boldsymbol{\mu}}^i_\Delta &= \text{MP}\left(\hat{\boldsymbol{\mu}}^i - \mu^i_f, \boldsymbol{x}^i - \mu^i_f\right), \quad \hat{\boldsymbol{\mu}}^i_* = \hat{\boldsymbol{\mu}}^i_\Delta + \mu^i_f.
\end{aligned} \quad (7)$$

Here, $\mu^i_f$ and $\sigma^i_f$ are the mean values of $\hat{\boldsymbol{\mu}}^i$ and $\hat{\boldsymbol{\sigma}}^i$. While $\hat{\boldsymbol{\mu}}^i_*$ and $\hat{\boldsymbol{\sigma}}^i_*$ denote the prediction of $\hat{\boldsymbol{\mu}}^i$ and $\hat{\boldsymbol{\sigma}}^i$. The MP is a mean prediction branch, and the SP is a standard deviation prediction branch. They are affiliated with DPM and adopt the same network structure.

**Time Domain Prediction** As the above frequency domain prediction process, for the distribution statistics of TDN, this prediction process can be formulated as follows:

$$\begin{aligned}
\boldsymbol{\sigma}^i_\Delta &= \text{SP}\left(\boldsymbol{\sigma}^i - \sigma^i_o, \boldsymbol{x}^i\right), \quad \boldsymbol{\sigma}^i_* = \boldsymbol{\sigma}^i_\Delta + \sigma^i_o, \\
\boldsymbol{\mu}^i_\Delta &= \text{MP}\left(\boldsymbol{\mu}^i - \mu^i_o, \boldsymbol{x}^i - \mu^i_o\right), \quad \boldsymbol{\mu}^i_* = \boldsymbol{\mu}^i_\Delta + \mu^i_o.
\end{aligned} \quad (8)$$

Here, $\mu^i_o$ and $\sigma^i_o$ are the mean values of $\boldsymbol{\mu}^i$ and $\boldsymbol{\sigma}^i$, respectively. Likewise, $\boldsymbol{\mu}^i_*$ and $\boldsymbol{\sigma}^i_*$ denote the prediction of $\boldsymbol{\mu}^i$ and $\boldsymbol{\sigma}^i$. The SP and the MP are noted in frequency domain prediction.

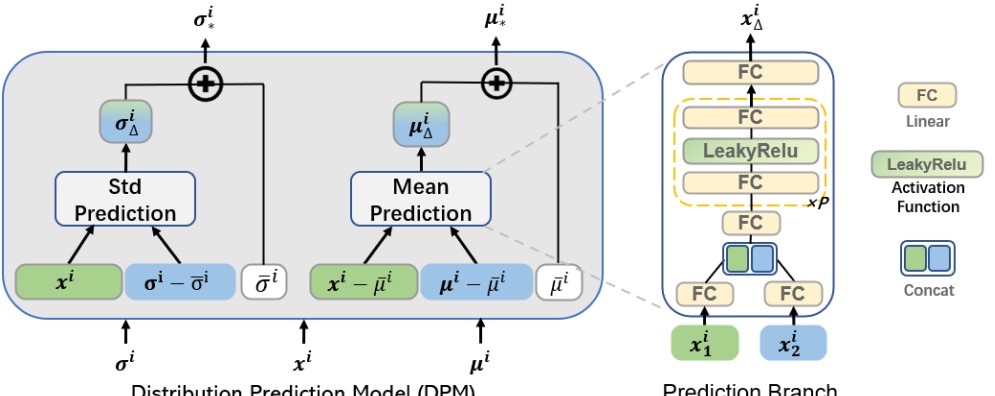

Figure 3: The architecture of the distribution prediction model primarily consists of two predictive branches: the Mean Prediction branch and the Standard Deviation (Std) Prediction branch. The specific network structure of these branches is illustrated in the Prediction Branch.

**De-normalization.** After the aforementioned distribution predictions, two sets of estimated distribution statistics will be gained. Which are weighted to reconstruct the non-stationary information of the output of the time series forecasting model. This process can be described as follows:

$$\boldsymbol{\mu}_*^i = \boldsymbol{\mu}_*^i \cdot \beta + \hat{\boldsymbol{\mu}}_*^i \cdot \alpha, \quad \boldsymbol{\sigma}_*^i = \boldsymbol{\sigma}_*^i \cdot \beta + \hat{\boldsymbol{\sigma}}_*^i \cdot \alpha, \quad \bar{\boldsymbol{x}}^i = \bar{\boldsymbol{x}}^i \cdot \beta + \hat{\boldsymbol{x}}^i \cdot \alpha.$$
$$\boldsymbol{y}_*^i = \bar{\boldsymbol{y}}_*^i \odot \left( \boldsymbol{\sigma}_*^i + \epsilon \right) + \boldsymbol{\mu}_*^i. \tag{9}$$

Where $\bar{\boldsymbol{y}}_*^i$ is the output of the time series forecasting model, and $\boldsymbol{y}_*^i$ represents the predicted sequence after reconstructing non-stationary information. While $\alpha$, $\beta$, and $\epsilon$ mentioned before.

## 3.4 Collaborative Training

Distribution prediction and future series forecasting are essentially a bi-level optimization problem [35, 36, 37], where distribution outputs significantly impact the future series output. To enhance the training effects of our models, we pre-train the DPM to yield a relatively well-trained DPM. This procedure can be formulated as follows:

$$\theta_d = \arg \min_{\text{MSE}} \left( \left( \boldsymbol{\mu}_*^i, \boldsymbol{\sigma}_*^i \right), \left( \boldsymbol{\mu}_y^i, \boldsymbol{\sigma}_y^i \right), \theta_d \right). \tag{10}$$

Where $\theta_d$ represents the parameters of the DPM, it is noteworthy that the wavelet bases $\phi_{l,h}$ and the weighted factor $\alpha$ belong to $\theta_d$. We select the mean square error (MSE) as our loss function between the predicted distribution and the ground truth of the distribution, acquired from the future sequence through TDN. Subsequently, assuming a total training duration of $T$ epochs, the parameters $\theta_d$ of the DPM will be frozen, then we train FM for $T_1$ epochs. Finally, DPM and FM will be subject to collaborative training during the remaining $T - T_1$ epochs. The process is as follows:

$$\theta_f = \arg \min_{\text{MSE}} \left( \boldsymbol{y}_*^i, \boldsymbol{y}_y^i, \theta_f \right), \quad \text{if } t \le T_1,$$
$$\{\theta_d, \theta_f\} = \arg \min_{\text{MSE}} \left( \boldsymbol{y}_*^i, \boldsymbol{y}_y^i, \{\theta_d, \theta_f\} \right), \quad \text{otherwise.} \tag{11}$$

Where $t$ denotes the $t^{th}$ epoch during the training process, and $\phi_f$ represents the parameters of the FM. We train DPM and FM concurrently to mitigate potential errors in the pretraining stage, as the ground truth of distribution is drived from future sequences based on distritution assumption. This ground truth is somewhat inconsistent with the actual situation and fails to account for high-frequency distribution changes. Consequently, we pretrain the DPM using assumption-based distribution ground truth, and then collaborative train it jointly based on the loss derived from the future sequences.

## 4 Experiments

In this section, we conduct comprehensive experiments on multiple real-world time series datasets to assess the effectiveness of our proposed reversible normalization method DDN.

**Datasets** We conduct extensive experiments on these seven popular real-world datasets [18], including **Electricity Transformer Temperature (ETT)** with its four subsets (ETTh1, ETTh2, ETTm1, ETTm2), **Weather**, **Electricity**, and **Traffic**. The setting of these datasets following original works [18, 20], and more descriptions about these datasets present in appendix A.1.

**Baselines.** DDN is a model-agnostic method that can be applied to any mainstream time series forecasting model. To demonstrate its versatility, we integrate DDN into several representative models, including the earlier proposed models **FEDformer** [19] and **Autoformer** [20], the CI model **DLinear** [27], and the CD model **iTransformer** [24].

**Implementation details.** Our experiments were conducted three times with a consistent random seed and averaged to mean values. The Mean Square Error (MSE) and Mean Absolute Error (MAE) are chosen as evaluation metrics, with MSE serving as the training loss. All models use the same prediction lengths $T = \{96, 192, 336, 720\}$. For the look-back window $L$, Autoformer [20] and FEDformer [19] use $L = 96$, while DLinear [27] and iTransformer [24] utilize $L = 336$ and $L = 720$ respectively. The wavelet bases initialize to the "coiflet" bases, the default size of our sliding window is set to 7 for information content and temporal locality balance, and $\alpha$ starts at zero. More implementation details of our experiments can be referred to appendix A.2.

## 4.1 Main Results

As illustrated, the DDN method significantly enhances the predictive performance of the four different baselines across nearly all datasets. For the MSE metric, this improvement is particularly evident in the three relatively large datasets: Weather, Electricity, and Traffic. Utilizing DDN, Autoformer achieves a relative error reduction of 19.2%, 24.7%, and 25.6%, respectively, while FEDformer achieves a relative error reduction of 13.1%, 16.2%, and 22.3%. Similarly, incorporating the DDN into the other models also results in substantial performance gains. Additionally, Autoformer, FEDformer, and DLinear do not employ reversible normalization in official implements. While iTransformer utilizes the RevIN [11] normalization technique based on static statistics. However, replacing the RevIN module in iTransformer with the DDN module still yields significant performance improvements. These results strongly demonstrate that DDN makes the baseline model more robust in forecasting.

| Methods | Autoformer | | +DDN | | FEDformer | | +DDN | | DLinear | | +DDN | | iTransformer | | +DDN | |
|---|---|---|---|---|---|---|---|---|---|---|---|---|---|---|---|---|
| Metric | MSE | MAE | MSE | MAE | MSE | MAE | MSE | MAE | MSE | MAE | MSE | MAE | MSE | MAE | MSE | MAE |
| ETTh1 96 | 0.458 | 0.448 | **0.427** | **0.424** | **0.371** | 0.411 | 0.385 | **0.408** | 0.377 | 0.399 | **0.372** | **0.396** | 0.392 | 0.422 | **0.377** | **0.405** |
| ETTh1 192 | 0.481 | 0.474 | **0.472** | **0.452** | **0.420** | 0.443 | 0.415 | 0.452 | 0.417 | 0.426 | **0.406** | **0.416** | 0.428 | 0.448 | **0.414** | **0.430** |
| ETTh1 336 | 0.508 | 0.485 | **0.498** | **0.466** | 0.446 | 0.459 | 0.458 | 0.452 | 0.464 | 0.461 | **0.432** | **0.434** | 0.467 | 0.475 | **0.453** | **0.456** |
| ETTh1 720 | 0.525 | 0.516 | **0.502** | **0.483** | 0.482 | 0.495 | 0.490 | **0.479** | 0.493 | 0.505 | **0.462** | **0.474** | 0.568 | 0.547 | **0.553** | **0.530** |
| ETTm1 96 | 0.493 | 0.470 | **0.354** | **0.390** | 0.362 | 0.408 | **0.313** | **0.364** | 0.301 | 0.344 | **0.288** | **0.342** | 0.322 | 0.371 | **0.301** | **0.355** |
| ETTm1 192 | 0.546 | 0.498 | **0.397** | **0.408** | 0.395 | 0.427 | **0.361** | **0.396** | 0.335 | 0.366 | **0.324** | **0.364** | 0.353 | 0.392 | **0.339** | **0.378** |
| ETTm1 336 | 0.658 | 0.543 | **0.429** | **0.433** | 0.441 | 0.454 | **0.417** | **0.430** | 0.370 | 0.387 | **0.356** | **0.385** | 0.385 | 0.410 | **0.370** | **0.396** |
| ETTm1 720 | 0.626 | 0.532 | **0.488** | **0.464** | 0.488 | 0.481 | **0.470** | **0.472** | 0.425 | 0.421 | **0.415** | **0.419** | 0.441 | 0.443 | **0.426** | **0.426** |
| Weather 96 | 0.247 | 0.320 | **0.190** | **0.243** | 0.246 | 0.328 | **0.174** | **0.237** | 0.175 | 0.237 | **0.146** | **0.201** | 0.177 | 0.228 | **0.148** | **0.210** |
| Weather 192 | 0.302 | 0.366 | **0.231** | **0.282** | 0.281 | 0.341 | **0.233** | **0.294** | 0.217 | 0.275 | **0.190** | **0.247** | 0.223 | 0.266 | **0.191** | **0.252** |
| Weather 336 | 0.362 | 0.394 | **0.289** | **0.327** | 0.337 | 0.376 | **0.307** | **0.349** | 0.263 | 0.314 | **0.239** | **0.288** | 0.287 | 0.310 | **0.237** | **0.290** |
| Weather 720 | 0.427 | 0.433 | **0.369** | **0.375** | 0.414 | 0.426 | **0.399** | **0.405** | 0.325 | 0.366 | **0.311** | **0.343** | 0.364 | 0.365 | **0.301** | **0.336** |
| Electricity 96 | 0.195 | 0.309 | **0.150** | **0.254** | 0.185 | 0.300 | **0.146** | **0.251** | 0.140 | 0.237 | **0.131** | **0.228** | 0.133 | 0.229 | **0.127** | **0.225** |
| Electricity 192 | 0.215 | 0.325 | **0.173** | **0.275** | 0.196 | 0.310 | **0.168** | **0.268** | 0.153 | 0.250 | **0.148** | **0.246** | 0.154 | 0.250 | **0.146** | **0.246** |
| Electricity 336 | 0.237 | 0.344 | **0.185** | **0.288** | 0.215 | 0.330 | **0.174** | **0.280** | 0.168 | 0.267 | **0.164** | **0.264** | 0.170 | 0.266 | **0.156** | **0.257** |
| Electricity 720 | 0.292 | 0.375 | **0.201** | **0.304** | 0.244 | 0.352 | **0.216** | **0.312** | 0.203 | 0.301 | **0.201** | **0.299** | 0.192 | 0.287 | **0.179** | **0.282** |
| Traffic 96 | 0.654 | 0.403 | **0.453** | **0.296** | 0.579 | 0.363 | **0.442** | **0.288** | 0.411 | 0.283 | **0.375** | **0.261** | 0.348 | 0.254 | **0.336** | **0.248** |
| Traffic 192 | 0.654 | 0.410 | **0.462** | **0.304** | 0.608 | 0.376 | **0.462** | **0.300** | 0.423 | 0.289 | **0.396** | **0.272** | 0.364 | 0.264 | **0.347** | **0.254** |
| Traffic 336 | 0.629 | 0.391 | **0.486** | **0.315** | 0.620 | 0.385 | **0.474** | **0.306** | 0.437 | 0.297 | **0.411** | **0.279** | 0.381 | 0.272 | **0.363** | **0.263** |
| Traffic 720 | 0.657 | 0.402 | **0.529** | **0.344** | 0.630 | 0.387 | **0.512** | **0.329** | 0.467 | 0.316 | **0.448** | **0.298** | 0.421 | 0.290 | **0.412** | **0.286** |

Table 2: Multivariate long-term forecasting results. The best results are highlighted in **bold**. More results can be found in Appendix D.1.

## 4.2 Comparison With Reversible Normalization Methods

**Quantitative evaluation.**    In this part, we compare recent representative reversible normalization methods using the average MSE metric across four prediction lengths in {96, 192, 336, 720}. Detailed descriptions of these methods and links for access are provided in the Appendix B. As shown in Table 3, DDN not only significantly enhances the predictive performance of the baseline models, including RevIN [11], NTS [13], and Dish-TS [12], but also demonstrates superior results when compared to existing reversible normalization methods. For instance, taking the recent slice-level SAN [14] method as an example, DDN achieves substantial improvements even upon its solid foundation. Further comparative details are detailed in Appendix D.2.

| Methods | Autoformer | | | | | | FEDformer | | | | | |
|---|---|---|---|---|---|---|---|---|---|---|---|---|
| | +DDN | +RevIN | +NST | +Dish-TS | +SAN | IMP | +DDN | +RevIN | +NST | +Dish-TS | +SAN | IMP |
| ETTh1 | **0.475** | 0.519 | 0.521 | 0.521 | 0.518 | 3.7% | **0.437** | 0.463 | 0.456 | 0.461 | 0.447 | -1.6% |
| ETTh2 | **0.403** | 0.489 | 0.465 | 1.175 | 0.411 | 9.6% | **0.385** | 0.465 | 0.481 | 1.004 | 0.404 | 9.8% |
| ETTm1 | **0.417** | 0.562 | 0.535 | 0.567 | 0.406 | 28.2% | 0.390 | 0.415 | 0.411 | 0.422 | **0.377** | 7.6% |
| ETTm2 | **0.283** | 0.325 | 0.331 | 0.894 | 0.311 | 15.0% | **0.282** | 0.310 | 0.315 | 0.759 | 0.287 | 6.6% |
| Weather | **0.270** | 0.290 | 0.290 | 0.433 | 0.305 | 19.2% | 0.278 | 0.268 | **0.267** | 0.398 | 0.279 | 13.1% |
| Electricity | **0.177** | 0.219 | 0.213 | 0.231 | 0.204 | 24.7% | **0.176** | 0.200 | 0.198 | 0.203 | 0.191 | 16.2% |
| Traffic | **0.483** | 0.666 | 0.664 | 0.677 | 0.594 | 25.6% | **0.473** | 0.647 | 0.649 | 0.652 | 0.572 | 22.3% |

Table 3: Comparison between DDN and existing reversible normalization methods of varying granularities. IMP represents the relative percentage improvement of DDN over the original sequence. The best results are highlighted in **bold**.

| Methods | DLinear | | | | iTransformer | | | |
|---|---|---|---|---|---|---|---|---|
| | +SAN | | +DDN | | +SAN | | +DDN | |
| Metric | MSE | MAE | MSE | MAE | MSE | MAE | MSE | MAE |
| Weather 96 | 0.152 | 0.210 | **0.146** | **0.201** | 0.150 | **0.208** | **0.148** | 0.210 |
| Weather 192 | 0.196 | 0.253 | **0.190** | **0.247** | 0.195 | 0.253 | **0.191** | **0.252** |
| Weather 336 | 0.246 | 0.296 | **0.239** | **0.288** | 0.248 | 0.295 | **0.237** | **0.290** |
| Weather 720 | 0.315 | 0.346 | **0.311** | **0.343** | 0.311 | 0.342 | **0.301** | **0.336** |
| Electricity 96 | 0.137 | 0.234 | **0.131** | **0.228** | 0.130 | 0.229 | **0.127** | **0.225** |
| Electricity 192 | 0.152 | 0.248 | **0.148** | **0.246** | 0.148 | 0.247 | **0.146** | **0.246** |
| Electricity 336 | 0.167 | **0.264** | **0.164** | 0.264 | 0.158 | 0.259 | **0.156** | **0.257** |
| Electricity 720 | 0.202 | **0.296** | **0.201** | 0.299 | 0.183 | 0.284 | **0.179** | **0.282** |
| Traffic 96 | 0.412 | 0.290 | **0.375** | **0.261** | 0.363 | 0.269 | **0.336** | **0.248** |
| Traffic 192 | 0.431 | 0.299 | **0.396** | **0.272** | 0.374 | 0.274 | **0.347** | **0.254** |
| Traffic 336 | 0.447 | 0.308 | **0.411** | **0.279** | 0.389 | 0.281 | **0.363** | **0.263** |
| Traffic 720 | 0.475 | 0.320 | **0.448** | **0.298** | 0.418 | 0.294 | **0.412** | **0.286** |

Table 4: Comparison of forecasting errors between the DDN and SAN. The best results are highlighted in **bold**.

Considering that early models often lacked robust generalizability as their naive modeling strategies, we additionally included comparisons with two recent representative models: DLinear [27] (modeling features with CI) and iTransformer [24] (modeling features with CD). These methods already have good non-stationary adaptability, so they can better reflect the performance upper limit of fine-grained normalization methods. As illustrated in table 4, DDN significantly outperforms the recent state-of-the-art model SAN in handling non-stationary information. Overall, DDN almost achieves the best performance compared to SAN in every forecasting case. Specifically, on the Traffic dataset, SAN [14] struggles to achieve satisfactory predictive performance and even performs worse than the original predictive model. In contrast, DDN demonstrates effects by a finer-grain non-stationarity processing.

**Qualitative Evaluation.**    As shown in the figure 4, we compare DDN with reversible normalization methods at different granularities. It can be observed that there are significant differences among the various reversible normalization methods, which are related to their specific implementations. From subplots (a) and (b), it is evident that RevIN [11] reconstructs non-stationary information based on the distribution characteristics of historical sequences, making the distribution of the predicted sequence closer to that of the historical sequence rather than future sequence. However, as significant

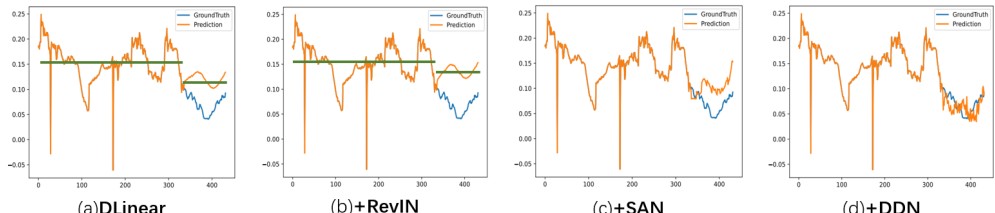

(a)**DLinear**    (b)**+RevIN**    (c)**+SAN**    (d)**+DDN**

Figure 4: Comparison of reversible normalization methods, samples from DLiner [27] weather dataset forecasting. Green solid lines represent the mean of the historical and predicted sequences.

distribution differences between historical and future sequences, this approach may even degrade the predictive accuracy. Comparing subplots (c) and (d), although slice level SAN [14] significantly improves overall predictive performance, it still lags behind point level DDN in terms of fine-grained variations. Additionally, it is worth highlighting that the fine-grained capability of DDN enables the reconstructed sequence to exhibit rapid local fluctuations flexibly. More comparisons can be found in Appendix C.

### 4.3  Dual-domain Dynamic Normalization Analysis

To validate the effectiveness of FDN and TDN, we conducted ablation studies comparing the predictive performance when only using FDN or TDN for non-stationary processing. It is important to note that when using FDN only for non-stationary processing, the ground truth of DPM pretraining comes from the non-stationary factors of FDN on future sequences. Correspondingly, wavelet bases will be set to nonlearnable parameters when we use FDN only. Meanwhile, for a fair comparison, both FDN and TDN in DDN utilize the same prediction model and share parameters, thereby avoiding the misconception that the superior performance of DDN stems from a larger parameter space in the prediction model.

The results of the ablation studies, as shown in Table 5, indicate that both FDN and TDN achieved outstanding performance, with FDN often reaching comparable or even superior predictive results compared to TDN alone. Furthermore, when we use FDN and TDN simultaneously, the predictive performance of DDN approaches even surpasses the best performance of FDN or TDN alone. It validates the effectiveness of TDN in capturing fine-grained non-stationarity at the point level in the time domain while confirming the robustness of FDN separating frequency components with different rapid changes in the frequency domain, thus enabling more refined non-stationary processing.

| Methods | | DLinear DDN | | TDN Only | | FDN Only | | iTransformer DDN | | TDN Only | | FDN Only | |
|---|---|---|---|---|---|---|---|---|---|---|---|---|---|
| Metric | | MSE | MAE | MSE | MAE | MSE | MAE | MSE | MAE | MSE | MAE | MSE | MAE |
| Weather | 96 | **0.146** | **0.201** | 0.150 | 0.203 | 0.149 | 0.206 | **0.148** | **0.210** | 0.153 | 0.212 | 0.153 | 0.216 |
| | 192 | **0.190** | **0.247** | 0.196 | 0.251 | 0.192 | 0.250 | **0.191** | **0.252** | 0.198 | 0.256 | 0.193 | 0.256 |
| | 336 | **0.239** | **0.288** | 0.247 | 0.294 | 0.242 | 0.293 | **0.237** | **0.290** | 0.249 | 0.302 | 0.245 | 0.305 |
| | 720 | **0.311** | **0.343** | 0.316 | 0.344 | 0.313 | 0.345 | **0.301** | **0.336** | 0.325 | 0.361 | 0.308 | 0.350 |
| Electricity | 96 | **0.131** | **0.228** | 0.133 | 0.230 | 0.132 | 0.229 | 0.127 | **0.225** | 0.130 | 0.227 | **0.126** | 0.226 |
| | 192 | **0.148** | **0.246** | 0.150 | 0.249 | 0.149 | **0.246** | 0.146 | 0.246 | 0.148 | 0.250 | **0.144** | **0.244** |
| | 336 | **0.164** | **0.264** | 0.165 | **0.264** | 0.166 | **0.264** | 0.156 | 0.257 | 0.160 | 0.259 | **0.156** | **0.257** |
| | 720 | 0.201 | 0.299 | 0.201 | **0.295** | **0.200** | 0.296 | **0.179** | **0.282** | 0.185 | 0.287 | 0.181 | 0.283 |
| Traffic | 96 | **0.375** | **0.261** | 0.378 | 0.263 | **0.375** | 0.263 | **0.336** | **0.248** | 0.338 | 0.250 | 0.338 | 0. 249 |
| | 192 | **0.396** | **0.272** | 0.399 | 0.278 | 0.298 | 0.274 | **0.347** | **0.254** | 0.352 | 0.257 | 0.351 | 0.256 |
| | 336 | **0.411** | **0.279** | 0.420 | 0.290 | 0.412 | 0.281 | **0.363** | **0.263** | 0.365 | 0.266 | 0.364 | **0.263** |
| | 720 | **0.448** | **0.298** | 0.460 | 0.310 | 0.450 | 0.300 | 0.412 | 0.286 | 0.418 | 0.288 | **0.408** | **0.282** |

Table 5: Ablation study of FDN and TDN. "TDN Only" and "FDN Only" indicate normalization using TDN only and FDN only, respectively. The best results are highlighted in **bold**.

## 5 Conclusion

In this work, we propose Dual-domain Dynamic Normalization (DDN), a novel method that dynamically captures non-stationary factors in time series forecasting, addressing sudden changes and distribution drifts in both time and frequency domains. Specifically, DDN employs sliding normalization in the time domain to eliminate and reconstruct non-stationary factors at a fine-grained level. In the frequency domain, it decomposes time series into high and low frequencies, effectively capturing rapid variations and sudden changes. As a model-agnostic auxiliary module, DDN significantly enhances the predictive performance of various forecasting models. Extensive experiments on seven real-world datasets validate the superiority of DDN, demonstrating its effectiveness in addressing distribution drift and improving the reliability of time series predictions.

## 6 Acknowledgments

This work is supported in part by the National Natural Science Foundation of China, under Grant (62302309, 62171248), Shenzhen Science and Technology Program (JCYJ20220818101014030, JCYJ20220818101012025), and the PCNL KEY project (PCL2023AS6-1).

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

# A    Experiments Setting

## A.1    Dataset Details

Our comprehensive experiments are conducted on seven time series datasets. Consistent with the methodologies of [18, 20, 24], we partition all datasets chronologically into training, validation, and testing subsets. Specifically, for the ETT dataset, we adopted a 6:2:2 split ratio, while a 7:1:2 ratio was utilized for the other datasets. Detailed descriptions of these datasets are as follows:

  (1) **ETT-small**[3] (Electricity Transformer Temperature) dataset: Comprises data from electricity transformers in two regions of China, collected between July 2016 and July 2018. It offers two different granularities: ETTh (1 hour) and ETTm (15 minutes). Each data point includes the value of oil temperature and six external power load features.

  (2) **Weather**[4] dataset: Comprises 21 distinct meteorological measurements in Germany, recorded every 10 minutes throughout 2020. It features key indicators such as air temperature and visibility, providing an in-depth view of weather patterns.

  (3) **Electricity**[5] dataset: Contains hourly electricity consumption data in kilowatt-hours (kWh) for 321 clients from 2012 to 2014, sourced from the UCI Machine Learning Repository.

  (4) **Traffic**[6] dataset: Features hourly data on road occupancy rates from 862 detectors in the San Francisco Bay area freeways, covering 2015 to 2016.

We provide access to all datasets through `https://github.com/thuml/iTransformer`. Detailed statistics for these datasets, including time steps, channels, and ADF test [38] results (evaluate the stationarity of a time series; a smaller value indicates greater non-stationarity.), are presented in Table A.1.

Table 6: The statistics of datasets.

| Datasets | Timesteps | Variates | Granularity | ADF |
|---|---|---|---|---|
| Electricity | 26304 | 321 | 1 hour | -8.44 |
| Weather | 52696 | 21 | 10 min | -26.68 |
| Traffic | 17544 | 862 | 1 hour | -15.02 |
| ETTh1 | 17420 | 7 | 1 hour | -5.91 |
| ETTh2 | 17420 | 7 | 1 hour | -4.13 |
| ETTm1 | 69680 | 7 | 15 min | -5.91 |
| ETTm2 | 69680 | 7 | 15 min | -5.66 |

## A.2    Setting Details

All experiments were conducted using PyTorch on a single NVIDIA 3090 24GB GPU. We utilize the ADAM optimizer [39] with an initial learning rate of $1e^{-4}$ for the distribution prediction model and employing Mean Squared Error (MSE) loss. The batch size, training epochs, and other baseline settings remain consistent with iTransformer [24]. The network for mean or standard deviation prediction comprises two feedforward Neural Network (FFN) layers, with dimensions of 512 for the first layer and 1024 for the second layer. We initialize the wavelet as Coiflet3, with $\alpha$ starting from 0. We conduct pre-training for 5 epochs and commence collaborative training from either the first or second epoch based on specific settings and datasets, aiming for improved training and model fitting.

---

[3] `https://github.com/zhouhaoyi/ETDataset`
[4] `https://www.bgc-jena.mpg.de/wetter/`
[5] `https://archive.ics.uci.edu/ml/datasets/ElectricityLoadDiagrams20112014`
[6] `https://pems.dot.ca.gov/`

### A.3 Baseline Methods

Our baseline methods are described as follows:

- Autoformer [20] is a transformer-based approach that adopts a decomposition strategy to learn complex temporal patterns in long-term prediction scenarios, decomposing time series into trend, cycle, and seasonal components, reflecting the long-term and seasonal aspects of the sequence data, respectively. The source code is available at https://github.com/thuml/Autoformer.

- FEDformer [19] is a hybrid Transformer-based model that integrates seasonal-trend decomposition and frequency enhancements. It can efficiently capture both global variations and intricate patterns. The source code is available at https://github.com/MAZiqing/FEDformer.

- DLinear [27] is a one-layer linear model challenging the efficacy of Transformer-based approaches in long-term time series forecasting, demonstrating superior performance on multiple datasets. The source code is available at https://github.com/cure-lab/LTSF-Linear.

- iTransformer [24] is a transformer-based model. The time series serve as variable tokens, utilizing self-attention mechanisms to capture correlations between multiple variables and using feedforward networks to encode the sequence representation. The source code is available at https://github.com/thuml/iTransformer.

## B  Reversible Normalization

Related reversible normalization methods are described in table 7, and they are described as follows:

- RevIN [11] introduces a data normalization method that addresses the limitations of simply eliminating non-stationary information, which can result in the loss of valuable data that the model needs to learn effectively. Unlike traditional methods that may lead to the model's inability to capture these critical non-stationary factors, this work proposes, for the first time, an explicit restoration of non-stationary information after the model's output. This ensures that while the model can learn without being affected by data drift, it also retains the essential non-stationary information. The source code is available at https://github.com/ts-kim/RevIN.

- NST [13] unlikes traditional time series forecasting methods that reduce non-stationarity by stabilizing the original data, this approach contradicts the importance of predicting sudden events in time series forecasting and overlooks the prevalence of non-stationary data in real-world scenarios, ultimately leading to overly stabilized modeling and prediction. To address this, this paper proposes a novel network architecture composed of sequence stabilization and inverse stabilization attention mechanisms. The source code is available at https://github.com/thuml/Nonstationary_Transformers.

- Dish-TS [12] notes that existing work on distribution shift in time series is often limited by distribution quantization and tends to overlook the potential distribution shift between the look back window and the horizon. To address this, this paper proposes using an MLP-based network to predict mean and standard deviation separately for the look back and horizon windows. The source code is available at https://github.com/weifantt/Dish-TS.

- SAN [14] indicates previous efforts on addressing non-stationarity have attempted to reduce it through normalization techniques. However, these methods typically overlook the distributional differences between input sequences and horizon sequences, assuming that all time points within the same instance share identical statistical properties. This overly idealistic approach can lead to suboptimal improvements. To address this issue, this paper introduces a novel slice-level adaptive normalization method. The source code is available at https://github.com/icantnamemyself/SAN.

Although NST [13] claims that non-stationary information can enhance feature diversity, previous methods have suffered from over-stationarization and inadequate consideration of non-stationary information utilization. However, it is noteworthy that, upon reviewing the current implementation

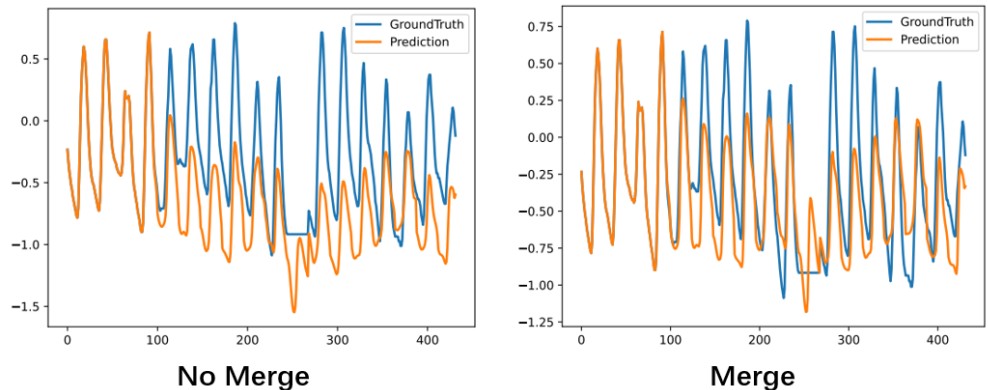

Figure 5: The comparison of NST [13], "Merge" means using non-stationary factors extraction module and merge to feature, 'No Merge" is the opposite.

| Method | Granularity | Estimation |
|---|---|---|
| RevIN [11] | Series Level | Statistics |
| NST [13] | Input/Output Level | Prediction |
| Dish-TS [12] | Input/Output Level | Prediction |
| SAN [14] | Slice Level | Prediction |
| **DDN** (Ours) | Point Level | Prediction |

Table 7: Comparative overview of non-stationary processing techniques in time series forecasting. "Granularity" and "Estimation" denote the normalization fineness and the prediction derivation method, respectively.

of the non-stationary transformer, the designed non-stationary information extraction module and the corresponding integration of this information into intermediate feature learning can essentially be seen as a mode that employs a distribution prediction network to estimate future non-stationary information and learns through the prediction network. Figure 5 presents a visual comparison of the effects with and without the integration of non-stationary information extraction. It is evident that the non-stationary information extraction and integration module fundamentally enables more accurate reconstruction of non-stationary information, such as mean value.

## B.1  Discrete Wavelet Transform

The Discrete Wavelet Transform (DWT) offers a nuanced approach to signal analysis by decomposing a time series into distinct frequency bands at multiple resolutions. This method is particularly adept at pinpointing both frequency and temporal aspects of a signal, making it invaluable for analyzing non-stationary time series. Initially, the general DWT of a time series $x(t)$ is expressed through the wavelet coefficients:

$$x_\phi(a, b) = \frac{1}{\sqrt{|a|}} \sum_t x(t) \cdot \phi\left(\frac{t - b}{a}\right) \tag{12}$$

where $a$ and $b$ denote the scaling and translation parameters, respectively, and $\phi$ is the mother wavelet function. Building upon this foundation, the DWT isolates the approximation coefficients (AC) and detail coefficients (DC), which capture distinct signal characteristics:

$$
\begin{aligned}
x_{ac}^i &= \text{DWT}_{\phi_{\text{low}}}(x^i) = \sum_t x(t) \cdot \phi_{\text{low}}(t), \\
x_{dc}^i &= \text{DWT}_{\phi_{\text{high}}}(x^i) = \sum_t x(t) \cdot \phi_{\text{high}}(t),
\end{aligned}
\tag{13}
$$

where $\phi_{\text{low}}(t)$ represents the low-pass filter and $\phi_{\text{high}}(t)$ the high-pass filter. Together, these filters facilitate the separation of the input time series into $x_{ac}^i$ and $x_{dc}^i$. The AC coefficients $x_{ac}^i$ embody the low-frequency components that outline the overarching trends within the time series, whereas the DC coefficients $x_{dc}^i$ encompass the high-frequency components, often associated with transient or noise elements in the signal.

The Inverse Discrete Wavelet Transform (IDWT) is then utilized to reconstruct the signal from its wavelet coefficients. The IDWT is the process that combines the AC and DC to form the original signal or an approximation of it. The reconstruction using IDWT can be written as:

$$x^i(t) = \text{IDWT}(x_{ac}^i, x_{dc}^i) = \sum_a x_{ac}^i(a) \cdot \phi_{ac}(a, t) + \sum_b x_{dc}^i(b) \cdot \phi_{dc}(b, t) \tag{14}$$

where $\phi_{ac}(a, t)$ and $\phi_{dc}(b, t)$ are the reconstruction functions from the approximation and detail coefficients, respectively. The wavelet transform, with its ability to localize both frequency and time, provides a powerful framework for signal analysis, particularly for signals like time series that contain non-stationary elements.

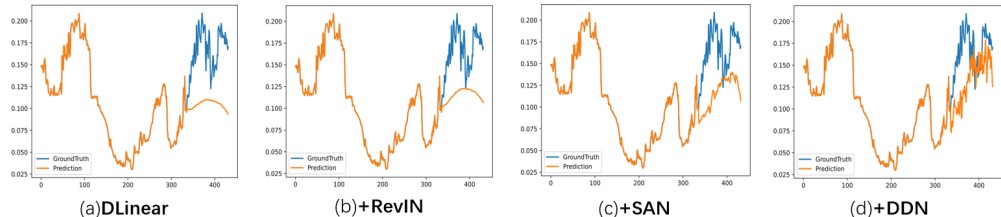

Figure 6: Comparison of reversible normalization methods, samples from DLiner [27] weather dataset forecasting.

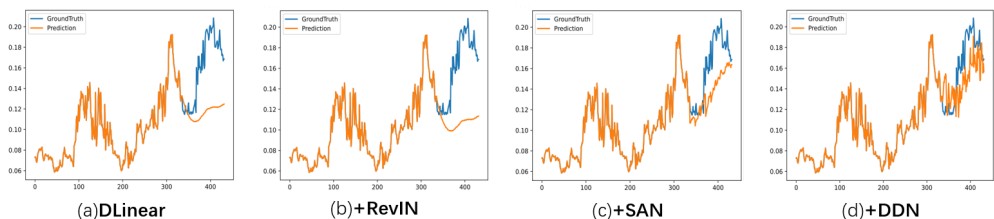

Figure 7: Comparison of reversible normalization methods, samples from DLiner [27] weather dataset forecasting.

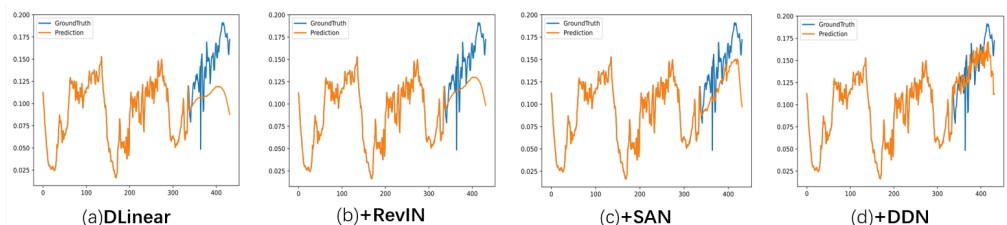

Figure 8: Comparison of reversible normalization methods, samples from DLiner [27] weather dataset forecasting.

## C   Visualization of Experiments

As illustrated in Figure 4, we present visual comparisons of predictions between different reversible normalization methods on the Weather dataset using DLinear [27]. The look-back window $L$ is 336, and the prediction length $T$ is 192. As discussed in Section 4.2, Figures 6, 7, and 8 highlight the necessity of a distribution change prediction module. Meanwhile, Figures 9 and 10 demonstrate the DDN method's superior capability in capturing details compared to the SAN method. This is particularly evident in the richer rapid changes present in its predictive output, allowing for dynamic adaptation and alignment with the rapidly changing data series in specific datasets.

To further substantiate this point, we conducted a visual comparison on the relatively smooth Traffic dataset in Figure 11, where these rapid changes also corresponded well with those in the original sequence, contrasting with the Weather dataset. This comparison underscores the enhanced detail and adaptability of the DDN method across different datasets.

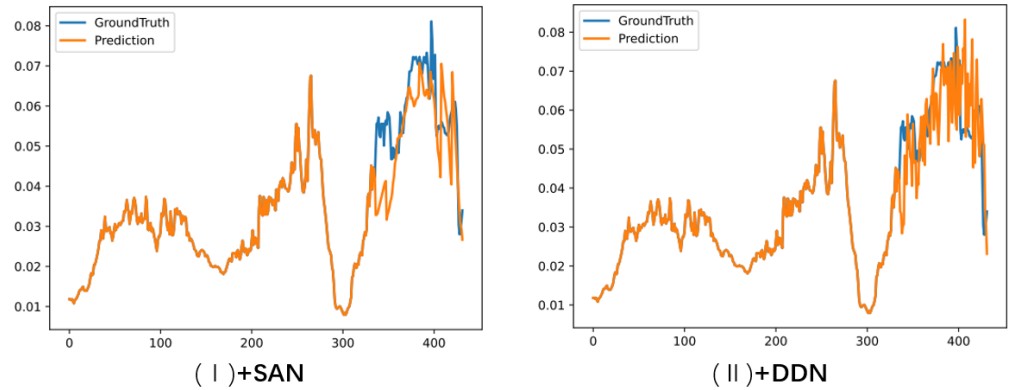

Figure 9: Comparison with slice level reversible normalization, samples from DLiner [27] weather dataset forecasting.

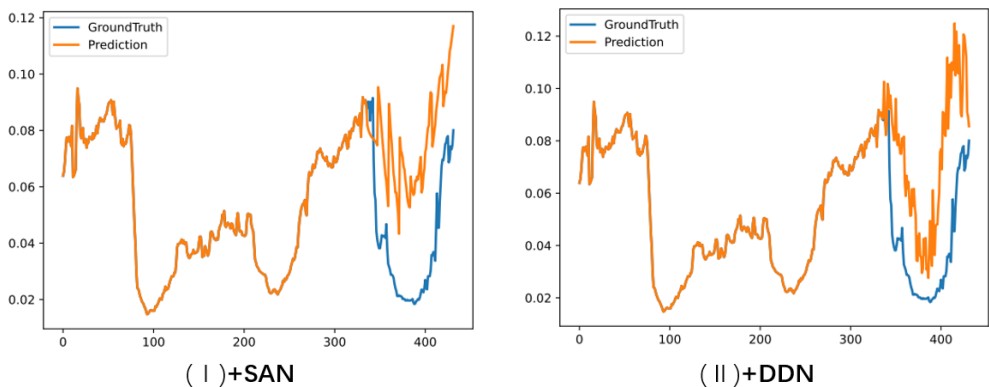

Figure 10: Comparison with slice level reversible normalization, samples from DLiner [27] weather dataset forecasting.

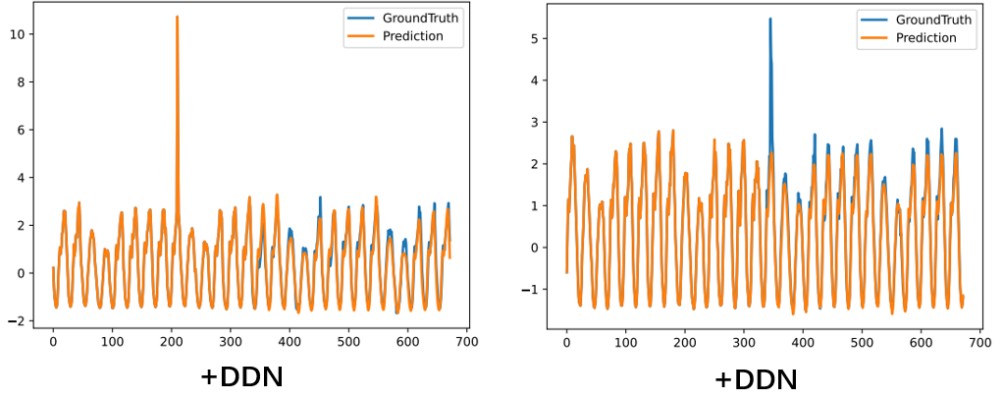

Figure 11: Our DDN reversible normalization method, samples from DLiner [27] Traffic dataset forecasting.

## D    Quantitative Evaluation Supplement

### D.1    Multivariable Forecasting Results of ETT

As illustrated in table 8, we present the comprehensive multivariate forecasting results on the ETT dataset in Table 5, encompassing the hourly datasets ETTh1 and ETTh2, as well as the 15-

minute datasets ETTm1 and ETTm2. The data clearly indicate that DDN demonstrates substantial enhancements across these datasets when applied to various backbone models.

| Methods | Autoformer | | +DDN | | FEDformer | | +DDN | | DLinear | | +DDN | | iTransformer | | +DDN | |
|---|---|---|---|---|---|---|---|---|---|---|---|---|---|---|---|---|
| Metric | MSE | MAE | MSE | MAE | MSE | MAE | MSE | MAE | MSE | MAE | MSE | MAE | MSE | MAE | MSE | MAE |
| ETTh1 96 | 0.458 | 0.448 | **0.427** | **0.424** | **0.371** | 0.411 | 0.385 | **0.408** | 0.377 | 0.399 | **0.372** | **0.396** | 0.392 | 0.422 | **0.377** | **0.405** |
| ETTh1 192 | 0.481 | 0.474 | **0.472** | **0.452** | **0.420** | 0.443 | 0.415 | **0.452** | 0.417 | 0.426 | **0.406** | **0.416** | 0.428 | 0.448 | **0.414** | **0.430** |
| ETTh1 336 | 0.508 | 0.485 | **0.498** | **0.466** | **0.446** | 0.459 | 0.458 | **0.452** | 0.464 | 0.461 | **0.432** | **0.434** | 0.467 | 0.475 | **0.453** | **0.456** |
| ETTh1 720 | 0.525 | 0.516 | **0.502** | **0.483** | **0.482** | 0.495 | 0.490 | **0.479** | 0.493 | 0.505 | **0.462** | **0.474** | 0.568 | 0.547 | **0.553** | **0.530** |
| ETTh2 96 | 0.384 | 0.420 | **0.350** | **0.385** | 0.341 | 0.382 | **0.312** | **0.357** | 0.292 | 0.356 | **0.279** | **0.340** | 0.315 | 0.366 | **0.279** | **0.342** |
| ETTh2 192 | 0.457 | 0.454 | **0.398** | **0.413** | 0.426 | 0.436 | **0.384** | **0.403** | 0.383 | 0.418 | **0.341** | **0.379** | 0.394 | 0.416 | **0.341** | **0.384** |
| ETTh2 336 | 0.468 | 0.473 | **0.428** | **0.444** | 0.481 | 0.479 | **0.421** | **0.437** | 0.473 | 0.477 | **0.364** | **0.402** | 0.430 | 0.445 | **0.369** | **0.410** |
| ETTh2 720 | 0.473 | 0.485 | **0.437** | **0.460** | 0.458 | 0.477 | **0.424** | **0.450** | 0.708 | 0.599 | **0.396** | **0.434** | 0.443 | 0.469 | **0.406** | **0.447** |
| ETTm1 96 | 0.493 | 0.470 | **0.354** | **0.390** | 0.362 | 0.408 | **0.313** | **0.364** | 0.301 | 0.344 | **0.288** | **0.342** | 0.322 | 0.371 | **0.301** | **0.355** |
| ETTm1 192 | 0.546 | 0.498 | **0.397** | **0.408** | 0.395 | 0.427 | **0.361** | **0.396** | 0.335 | 0.366 | **0.324** | **0.364** | 0.353 | 0.392 | **0.339** | **0.378** |
| ETTm1 336 | 0.658 | 0.543 | **0.429** | **0.433** | 0.441 | 0.454 | **0.417** | **0.430** | 0.370 | 0.387 | **0.356** | **0.385** | 0.385 | 0.410 | **0.370** | **0.396** |
| ETTm1 720 | 0.626 | 0.532 | **0.488** | **0.464** | 0.488 | 0.481 | **0.470** | **0.472** | 0.425 | 0.421 | **0.415** | **0.419** | 0.441 | 0.443 | **0.426** | **0.426** |
| ETTm2 96 | 0.261 | 0.329 | **0.177** | **0.262** | 0.191 | 0.283 | **0.171** | **0.255** | 0.169 | 0.263 | **0.167** | **0.257** | 0.187 | 0.279 | **0.162** | **0.253** |
| ETTm2 192 | 0.282 | 0.339 | **0.240** | **0.304** | 0.261 | 0.326 | **0.240** | **0.298** | 0.232 | 0.310 | **0.225** | **0.298** | 0.246 | 0.318 | **0.217** | **0.291** |
| ETTm2 336 | 0.350 | 0.378 | **0.306** | **0.344** | 0.327 | 0.365 | **0.306** | **0.342** | 0.303 | 0.361 | **0.286** | **0.339** | 0.300 | 0.354 | **0.269** | **0.327** |
| ETTm2 720 | 0.438 | 0.428 | **0.409** | **0.421** | 0.428 | 0.423 | **0.413** | **0.410** | 0.403 | 0.424 | **0.371** | **0.391** | 0.378 | 0.403 | **0.350** | **0.380** |

Table 8: Forecasting results comparison under different prediction lengths $T \in \{96, 192, 336, 720\}$ on ETT dataset. The input sequence length $L = 96$ for Autoformer and FEDformer, $L = 336$ for DLinear, and $L = 720$ for iTransformer. The **bold** values indicate best performance.

## D.2 Comparison between DDN and Reversible Normalization Methods

This section comprehensively compares DDN with existing reversible normalization methods. As shown in Table 9, our method consistently achieves state-of-the-art performance across almost all datasets, with a slight under-performance on the ETTm1 dataset compared to SAN. This demonstrates the versatility and effectiveness of our approach.

## D.3 Collaborative Training Ablation

Although recent works have shown that the two-stage training strategy effectively enhances the training of distribution prediction models, this training overly relies on the distribution ground truth obtained through distribution assumptions, which often contain inaccuracies. To address this, we introduce a collaborative training strategy that adjusts the distribution prediction model using the MSE loss between the actual sequence and the prediction during training. As illustrated in table 10, collaborative training generally yields superior training outcomes, and its lower bound in performance is comparable to that of the two-stage training strategy. This indicates that collaborative training can better mitigate the errors introduced by distribution assumptions and improve the accuracy of distribution prediction models.

## D.4 Univariate Forecasting Results

Following the same settings as our main experiment, we present the univariate forecasting results of Autoformer [20] and FEDformer [19] across three datasets, including Weather, Electricity, and Traffic, in Table 11. Similar to the multivariate forecasting results, DDN consistently enhances the performance of mainstream forecasting models in nearly all cases. It demonstrates that DDN is applicable in both multivariate time series forecasting and univariate forecasting.

| | | Autoformer | | | | | | | | | | FEDformer | | | | | | | | | |
|---|---|---|---|---|---|---|---|---|---|---|---|---|---|---|---|---|---|---|---|---|---|
| Methods | | +DDN | | +RevIN | | +NST | | +Dish-TS | | +SAN | | +DDN | | +RevIN | | +NST | | +Dish-TS | | +SAN | |
| Metric | | MSE | MAE | MSE | MAE | MSE | MAE | MSE | MAE | MSE | MAE | MSE | MAE | MSE | MAE | MSE | MAE | MSE | MAE | MSE | MAE |
| ETTh1 | 96 | **0.427** | **0.424** | 0.491 | 0.463 | 0.550 | 0.503 | 0.456 | 0.454 | 0.522 | 0.474 | **0.385** | **0.408** | 0.392 | 0.413 | 0.394 | 0.414 | 0.390 | 0.424 | 0.383 | 0.409 |
| | 192 | **0.472** | **0.452** | 0.513 | 0.478 | 0.530 | 0.492 | 0.495 | 0.480 | 0.498 | 0.472 | **0.415** | **0.452** | 0.443 | 0.444 | 0.441 | 0.442 | 0.441 | 0.458 | 0.431 | 0.438 |
| | 336 | **0.498** | **0.466** | 0.528 | 0.485 | 0.524 | 0.484 | 0.539 | 0.496 | 0.571 | 0.509 | **0.458** | **0.452** | 0.495 | 0.467 | 0.485 | 0.466 | 0.495 | 0.486 | 0.471 | 0.456 |
| | 720 | **0.502** | **0.483** | 0.543 | 0.510 | 0.510 | 0.491 | 0.563 | 0.522 | 0.555 | 0.514 | **0.490** | **0.479** | 0.520 | 0.498 | 0.505 | 0.496 | 0.519 | 0.509 | 0.504 | 0.488 |
| ETTh2 | 96 | 0.350 | 0.385 | 0.411 | 0.410 | 0.394 | 0.398 | 1.100 | 0.670 | **0.316** | **0.366** | **0.312** | **0.357** | 0.380 | 0.402 | 0.381 | 0.403 | 0.806 | 0.589 | 0.300 | 0.355 |
| | 192 | **0.398** | **0.413** | 0.478 | 0.450 | 0.473 | 0.450 | 0.976 | 0.672 | 0.413 | 0.426 | **0.384** | **0.403** | 0.457 | 0.433 | 0.478 | 0.453 | 0.936 | 0.659 | 0.392 | 0.413 |
| | 336 | **0.428** | **0.444** | 0.545 | 0.493 | 0.528 | 0.490 | 1.521 | 0.783 | 0.446 | 0.457 | **0.421** | **0.437** | 0.515 | 0.479 | 0.561 | 0.499 | 1.039 | 0.702 | 0.459 | 0.462 |
| | 720 | **0.437** | **0.460** | 0.523 | 0.490 | 0.524 | 0.498 | 1.105 | 0.745 | 0.471 | 0.474 | **0.424** | **0.450** | 0.507 | 0.487 | 0.502 | 0.481 | 1.237 | 0.759 | 0.462 | 0.472 |
| ETTm1 | 96 | 0.354 | 0.390 | 0.458 | 0.446 | 0.468 | 0.448 | 0.477 | 0.460 | **0.343** | **0.378** | 0.313 | 0.364 | 0.340 | 0.385 | 0.336 | 0.382 | 0.348 | 0.397 | **0.311** | **0.355** |
| | 192 | 0.397 | 0.408 | 0.560 | 0.491 | 0.526 | 0.468 | 0.545 | 0.488 | **0.390** | **0.400** | 0.361 | 0.396 | 0.390 | 0.411 | 0.386 | 0.409 | 0.406 | 0.428 | **0.351** | **0.383** |
| | 336 | 0.429 | 0.433 | 0.607 | 0.508 | 0.786 | 0.559 | 0.650 | 0.533 | **0.415** | **0.418** | 0.417 | 0.430 | 0.432 | 0.436 | 0.438 | 0.441 | 0.438 | 0.450 | **0.390** | **0.407** |
| | 720 | 0.488 | 0.464 | 0.623 | 0.526 | 0.564 | 0.501 | 0.595 | 0.518 | **0.476** | **0.453** | 0.470 | 0.472 | 0.497 | 0.466 | 0.483 | 0.460 | 0.497 | 0.481 | **0.456** | **0.444** |
| ETTm2 | 96 | **0.177** | **0.262** | 0.233 | 0.307 | 0.253 | 0.323 | 0.976 | 0.57 | 0.236 | 0.317 | **0.171** | **0.255** | 0.192 | 0.272 | 0.191 | 0.272 | 0.394 | 0.395 | 0.175 | 0.266 |
| | 192 | **0.240** | **0.304** | 0.288 | 0.337 | 0.289 | 0.335 | 0.532 | 0.485 | 0.260 | 0.329 | **0.240** | **0.298** | 0.270 | 0.320 | 0.270 | 0.321 | 0.552 | 0.472 | 0.246 | 0.315 |
| | 336 | **0.306** | **0.344** | 0.345 | 0.370 | 0.339 | 0.365 | 0.795 | 0.592 | 0.330 | 0.376 | **0.306** | **0.342** | 0.348 | 0.367 | 0.353 | 0.371 | 0.808 | 0.601 | 0.315 | 0.362 |
| | 720 | **0.409** | **0.421** | 0.434 | 0.419 | 0.426 | 0.432 | 1.271 | 0.768 | 0.417 | 0.428 | **0.413** | **0.410** | 0.430 | 0.415 | 0.445 | 0.422 | 1.282 | 0.771 | 0.412 | 0.422 |
| Weather | 96 | **0.190** | **0.243** | 0.212 | 0.257 | 0.211 | 0.254 | 0.268 | 0.338 | 0.194 | 0.256 | **0.174** | **0.237** | 0.187 | 0.234 | 0.187 | 0.234 | 0.244 | 0.317 | 0.179 | 0.239 |
| | 192 | **0.231** | **0.282** | 0.264 | 0.300 | 0.265 | 0.301 | 0.376 | 0.421 | 0.258 | 0.316 | **0.233** | **0.294** | 0.235 | 0.272 | 0.235 | 0.272 | 0.320 | 0.980 | 0.234 | 0.296 |
| | 336 | **0.289** | **0.327** | 0.309 | 0.329 | 0.303 | 0.324 | 0.475 | 0.486 | 0.329 | 0.367 | 0.307 | 0.349 | **0.287** | **0.307** | 0.289 | 0.308 | 0.424 | 0.502 | 0.304 | 0.384 |
| | 720 | 0.369 | 0.375 | 0.377 | 0.367 | 0.366 | **0.357** | 0.612 | 0.560 | 0.440 | 0.438 | 0.399 | 0.405 | 0.361 | 0.353 | **0.359** | **0.352** | 0.604 | 0.553 | 0.400 | 0.404 |
| Electricity | 96 | **0.150** | **0.254** | 0.179 | 0.286 | 0.179 | 0.285 | 0.197 | 0.290 | 0.172 | 0.281 | **0.146** | **0.251** | 0.172 | 0.278 | 0.172 | 0.279 | 0.175 | 0.284 | 0.164 | 0.272 |
| | 192 | **0.173** | **0.275** | 0.216 | 0.316 | 0.209 | 0.309 | 0.215 | 0.318 | 0.195 | 0.300 | **0.168** | **0.268** | 0.185 | 0.289 | 0.187 | 0.291 | 0.188 | 0.296 | 0.179 | 0.286 |
| | 336 | **0.185** | **0.288** | 0.233 | 0.331 | 0.246 | 0.335 | 0.244 | 0.343 | 0.211 | 0.316 | **0.174** | **0.280** | 0.200 | 0.304 | 0.202 | 0.307 | 0.209 | 0.319 | 0.191 | 0.299 |
| | 720 | **0.201** | **0.304** | 0.246 | 0.341 | 0.252 | 0.354 | 0.286 | 0.370 | 0.236 | 0.335 | **0.216** | **0.312** | 0.243 | 0.337 | 0.230 | 0.326 | 0.239 | 0.343 | 0.230 | 0.334 |
| Traffic | 96 | **0.453** | **0.296** | 0.643 | 0.354 | 0.654 | 0.354 | 0.652 | 0.363 | 0.569 | 0.350 | **0.422** | **0.288** | 0.613 | 0.347 | 0.612 | 0.348 | 0.613 | 0.350 | 0.536 | 0.330 |
| | 192 | **0.462** | **0.304** | 0.659 | 0.373 | 0.643 | 0.367 | 0.669 | 0.374 | 0.594 | 0.364 | **0.462** | **0.300** | 0.637 | 0.356 | 0.641 | 0.357 | 0.644 | 0.362 | 0.565 | 0.345 |
| | 336 | **0.486** | **0.315** | 0.662 | 0.371 | 0.665 | 0.363 | 0.683 | 0.376 | 0.591 | 0.363 | **0.474** | **0.306** | 0.652 | 0.363 | 0.654 | 0.363 | 0.659 | 0.370 | 0.580 | 0.354 |
| | 720 | **0.529** | **0.344** | 0.700 | 0.384 | 0.667 | 0.373 | 0.703 | 0.392 | 0.623 | 0.380 | **0.512** | **0.329** | 0.686 | 0.382 | 0.688 | 0.380 | 0.693 | 0.388 | 0.607 | 0.367 |

Table 9: Comparison of forecasting errors between different reversible normalization methods. The **bold** values indicate best performance.

| | | DLinear | | | | iTransformer | | | |
|---|---|---|---|---|---|---|---|---|---|
| Methods | | Co-train | | Wo Co-train | | Co-train | | Wo Co-train | |
| Metric | | MSE | MAE | MSE | MAE | MSE | MAE | MSE | MAE |
| Weather | 96 | **0.146** | **0.201** | 0.151 | 0.209 | **0.148** | **0.210** | 0.149 | 0.212 |
| | 192 | **0.190** | **0.247** | 0.193 | 0.251 | **0.191** | **0.252** | 0.191 | 0.252 |
| | 336 | **0.239** | **0.288** | 0.241 | 0.291 | **0.237** | 0.290 | 0.236 | 0.289 |
| | 720 | **0.311** | **0.343** | 0.308 | 0.343 | **0.301** | **0.336** | 0.301 | 0.337 |
| Electricity | 96 | **0.131** | **0.228** | 0.135 | 0.231 | **0.127** | **0.225** | 0.130 | 0.229 |
| | 192 | **0.148** | **0.246** | 0.150 | 0.247 | **0.146** | **0.246** | 0.147 | **0.246** |
| | 336 | **0.164** | **0.264** | 0.167 | 0.266 | **0.156** | **0.257** | 0.157 | 0.258 |
| | 720 | **0.201** | **0.299** | 0.210 | 0.309 | **0.179** | **0.282** | 0.184 | 0.286 |
| Traffic | 96 | **0.375** | **0.261** | 0.395 | 0.279 | **0.336** | **0.248** | 0.344 | 0.254 |
| | 192 | **0.396** | **0.272** | 0.412 | 0.286 | **0.347** | **0.254** | 0.354 | 0.261 |
| | 336 | **0.411** | **0.279** | 0.425 | 0.291 | **0.363** | **0.263** | 0.370 | 0.268 |
| | 720 | **0.448** | **0.298** | 0.452 | 0.308 | **0.412** | **0.286** | 0.410 | 0.288 |

Table 10: Comparison between the collaborative training strategy and the conventional two-stage training strategy. "Co-train" denotes the use of the collaborative training strategy, while "Wo Co-train" signifies the approach where the distribution prediction model's parameters are frozen after pre-training, and only the time series forecasting model is trained. The **bold** values indicate best performance.

| Methods | Autoformer | | +DDN | | FEDformer | | +DDN | |
|---|---|---|---|---|---|---|---|---|
| Metric | MSE | MAE | MSE | MAE | MSE | MAE | MSE | MAE |
| **Weather** 96 | 0.004 | 0.047 | **0.003** | **0.044** | **0.002** | **0.037** | 0.003 | 0.040 |
| 192 | 0.003 | 0.045 | **0.003** | **0.039** | 0.005 | 0.058 | **0.004** | **0.052** |
| 336 | 0.008 | 0.068 | **0.003** | **0.041** | 0.003 | 0.045 | **0.002** | **0.031** |
| 720 | 0.058 | 0.176 | **0.003** | **0.044** | 0.011 | 0.080 | **0.004** | **0.052** |
| **Electricity** 96 | 0.442 | 0.490 | **0.346** | **0.424** | 0.302 | 0.413 | **0.213** | **0.330** |
| 192 | 0.555 | 0.550 | **0.342** | **0.423** | 0.377 | 0.459 | **0.252** | **0.357** |
| 336 | 0.617 | 0.620 | **0.380** | **0.447** | 0.673 | 0.636 | **0.313** | **0.401** |
| 720 | 0.645 | 0.624 | **0.463** | **0.505** | 0.575 | 0.575 | **0.410** | **0.454** |
| **Traffic** 96 | 0.265 | 0.375 | **0.153** | **0.239** | 0.179 | 0.282 | **0.137** | **0.217** |
| 192 | 0.266 | 0.372 | **0.156** | **0.242** | 0.211 | 0.316 | **0.142** | **0.220** |
| 336 | 0.284 | 0.371 | **0.164** | **0.256** | 0.369 | 0.458 | **0.140** | **0.222** |
| 720 | 0.260 | 0.369 | **0.187** | **0.279** | 0.300 | 0.407 | **0.157** | **0.242** |

Table 11: Univariate forecasting results. The **bold** values indicate best performance.

