# OpenReview forum: "DDN: Dual-domain Dynamic Normalization for Non-stationary Time Series Forecasting"
_NeurIPS.cc/2024/Conference — NeurIPS 2024 poster_

### Official Review · Reviewer_G9hB · 2024-06-15

**Soundness:** 3
**Presentation:** 2
**Contribution:** 3
**Rating:** 5
**Confidence:** 5

**Summary:**

This paper proposes a novel Dual-domain Dynamic Normalization (DDN) method to dynamically capture distribution variations in both time and frequency domains, leveraging wavelet transform to handle time-varying periods. DDN eliminates non-stationarity in time series through normalization within a sliding window and can be easily integrated into various forecasting models as a plug-and-play module. Extensive experiments on public benchmark datasets demonstrate DDN's superior performance over existing normalization methods. The code will be made available after the review process.

**Strengths:**

1. The experimental results seem promising.

2. This study combines existing techniques in a new way.

**Weaknesses:**

1. The writing quality and presentation require necessary improvements. I have identified some obvious typos and errors upon a quick browse of the article:

- Line 119: The period should be replaced by a comma, while the comma should be replaced by a period.

- Line 167: The period should be replaced by a comma.

- Equation 10 and Equation 11: The notation below "argmin" should define the domain or the set of possible values for the variables being considered for minimization. Additionally, Equation 10 should be followed by a comma, as should Equation 11.


2. It is challenging to discern the distinction of their method from Figure 1, as the resulting outcomes of the existing manipulation and the proposed manipulation appear very similar. The authors should consider a better presentation to highlight the differences between their method and existing ones.


3. The description of some key concepts needs refinement to reduce unclarity and ambiguity. The authors challenge previous studies for “a fixed window” as stated in line 37. Before reviewing the entire method, this phrase might be misunderstood as referring to “a fixed window size,” suggesting that the primary motivation of this study is to adaptively adjust the window size. The subsequent expression “time-varying period” also leads to potential confusion. The authors should carefully consider how to present their core ideas with greater accuracy.


4. The implementation and core idea of SlideNorm bear significant resemblance to a method proposed by an existing study [1], necessitating a comparison and clarification of its distinction.

[1] J. Deng et al., "Disentangling Structured Components: Towards Adaptive, Interpretable and Scalable Time Series Forecasting," in IEEE Transactions on Knowledge and Data Engineering, doi: 10.1109/TKDE.2024.3371931.


5. The realm of probabilistic time series forecasting also targets the same task of predicting the distribution of future data with DPM. Demonstrating that DPM surpasses some representative probabilistic forecasting models would make this study more solid and convincing. Furthermore, the authors should justify that the improvements achieved by their method arise from the estimation of data distribution rather than the combined power of two deep learning models.


6. The evaluation is somewhat limited. Assessing effectiveness against different settings of hyper-parameters, such as sliding window size and look-back window size, can enhance this section.

**Questions:**

Please address the weak points above.

**Limitations:**

Neither limitations nor broader impacts are discussed at the end of this paper. As

---

> ### Author Rebuttal · Authors · 2024-08-07
>
> Thanks for your valuable comments of our work, we will carefully review and revise our manuscript to correct any inappropriate expressions or symbol errors. Here are responses to your concerns and questions:
>
> **Question 1:** The implementation and core idea of SlideNorm bear significant resemblance to a method proposed by an existing study [1], necessitating a comparison and clarification of its distinction.
>
> **Response 1:** We are pleased to find several conceptual similarities between existing study and DDN, such as:
>
> 1. Calculating the distribution characteristics by sliding windows.
>
> 2. Employing a large sliding window to obtain long-term changes with minimal bias and eliminate local high-frequency fluctuations, and a small sliding window aims at short-term rapid changes.
>
> The main differences between two works are as follows:
>
> **Target:** DDN is a plugin module for distribution prediction, and SCNN [1] is a model for future forecasting. DDN eliminates the non-stationary factors of the **input series** and reconstructs them for the **output series**. SCNN separates non-stationary factors from **intermediate representation**. Thus, DDN can be better compatible with existing models and replace their reversible normalization module (RevIN) without considering intermediate features.
>
> **Experiment:** To validate the above statement, we replaced the original reversible normalization module in SCNN with DDN. The training epoch will only be modified from 200 to 50 due to time constraints. *The OOM is out of memory*.
> |||||||
> |-|-|-|-|-|-|
> |L=168|SCNN|SCNN|DDN|DDN|
> |Metric|MSE|MAE|MSE|MAE|
> |ELC|||||
> |96|0.145|0.238|0.136|0.233|
> |192|0.160|0.252|0.156|0.250|
> |336|0.177|0.267|0.169|0.263|
> |720|0.219|0.303|0.215|0.301|
> |Traffic|||||
> |96|0.386|0.271|0.384|0.270|
> |192|0.416|0.280|0.412|0.275|
> |336|0.435|0.285|0.429|0.281|
> |720|*OOM*|*OOM*|*OOM*|*OOM*|
>
> **Distribution Calculation:**
> SCNN directly applies sliding windows of different sizes to intermediate features to obtain various statistics for long-term low-frequency and short-term high-frequency components. DDN separates high and low-frequency components by Wavelet Transform. Then employing a large sliding window on low-frequency, obtaining long-term changes with minimal bias and **eliminate local high-frequency fluctuations**. A small sliding window on high-frequency component aims at **short-term rapid changes**.
>
> **Training strategy:** SCNN sets distribution characteristics as a penalty term in the loss function. In contrast, DDN builds upon the two-stage training strategy [2, 3], which means DDN does not need to modify the original loss function of baseline.
> Question 2: Demonstrating that DPM surpasses some representative probabilistic forecasting models would make this study more solid and convincing. The authors should justify that the improvements achieved by their method arise from the estimation of data distribution rather than the combined power of two deep learning models.
>
> **Response 2:**
> 1. DPM predicts distribution, not future sequence. Probabilistic time series forecasting incorporates uncertainty into prediction, allowing for a more flexible description of the future sequence distribution. However, the small scale and noise of time series datasets pose challenges for probabilistic methods to provide ideal predictive performance. Meanwhile, the long computational time limits their application as a plugin. Therefore, current DPM typically uses deterministic methods based on MLP [2, 3] rather than probabilistic methods.
>
> 2. Assessing the effectiveness of distribution prediction is challenging, as the true ground truth of data distribution is unattainable. Therefore, we designed a new experiment to evaluate and validate distribution prediction performance. To ensure DPM is used for **distribution prediction rather than directly predicting future series**, we first train DPM to predict mean and standard deviation with distributional loss, which is calculated from the DPM outputs and the distribution extracted from horizontal series. Then, we freeze the DPM and apply it to existing models. The experiment is as follows:
> ||||||||||
> |-|-|-|-|-|-|-|-|-|
> ||DLinear|DLinear|+DDN|+DDN|iTransformer|iTransformer|+DDN|+DDN|
> |Metric|MSE|MAE|MSE|MAE|MSE|MAE|MSE|MAE|
> |Weather|0.245|0.298|0.223|0.274|0.263|0.292|0.219|0.273|
> |ELC|0.166|0.264|0.165|0.263|0.162|0.258|0.155|0.255|
> |Traffic|0.435|0.296|0.421|0.291|0.379|0.27|0.37|0.268|
>
> We provide the average results for prediction lengths of {96, 192, 336, 720}. These results demonstrate that the distribution prediction capability of DPM enhances the performance of existing models. Additionally, it is worthy to note that recent works [2, 3] also combine DPM with existing models, but they often show performance degradation on Weather, Electricity, and Traffic. For instance, on the Traffic dataset with SAN, the MSE metric for DLinear and iTransformer decreased by 1.53% and 1.94%, respectively.
>
> **Question 3:** Limited hyperparameter sensitivity analysis, such as sliding window size and look-back window size.
>
> **Response 3:** Please refer to Reviewer i5Qt's Response 1 and 3.
>
> [1] J. Deng et al., "Disentangling Structured Components: Towards Adaptive, Interpretable and Scalable Time Series Forecasting," in IEEE Transactions on Knowledge and Data Engineering, doi: 10.1109/TKDE.2024.3371931.
>
> [2] Liu Z, Cheng M, Li Z, et al. Adaptive normalization for non-stationary time series forecasting: A temporal slice perspective[J]. Advances in Neural Information Processing Systems, 2024, 36.
>
> [3] Han L, Ye H J, Zhan D C. SIN: Selective and Interpretable Normalization for Long-Term Time Series Forecasting[C]//Forty-first International Conference on Machine Learning.

---

> > ### Comment · Reviewer_G9hB · 2024-08-09
> >
> > I appreciate that the authors have addressed some of my concerns, and as a result, I am inclined to raise my score to 5.

---

> > > ### Author Response · Authors · 2024-08-11
> > > **Response to reviewer**
> > >
> > > Thanks a lot for your valuable comments for our work.  If you have any additional question, we are happy to answer any additional question before the rebuttal ends.

---

### Official Review · Reviewer_qBzF · 2024-07-08

**Soundness:** 4
**Presentation:** 3
**Contribution:** 3
**Rating:** 7
**Confidence:** 5

**Summary:**

This paper proposes a novel Dual-Domain Dynamic Normalization (DDN) method to address the non-stationary variations in real-world time series by operating in both the time and frequency domains. In the frequency domain, wavelet transform is employed to decompose the time series into high and low-frequency components to capture distributional changes across different frequency scales. In the time domain, sliding statistics are introduced to adapt to the rapid changes of non-stationary data. As a plug-and-play module, this method can be easily integrated into existing predictive models. Experimental results demonstrate that DDN significantly enhances the predictive performance of existing models.

**Strengths:**

1. The paper presents a highly innovative and effective method. By leveraging the inherent differences in variation between high and low-frequency components, it introduces a separation method to capture distributional changes at different frequency scales. Meanwhile, the paper introduces the concept of sliding statistics (rolling statistics) to dynamically capture distributional changes over time.

2. Benefiting from the dynamic nature of time-domain operations and the complementary multi-frequency scale characteristics of frequency-domain operations, the proposed method effectively identifies distributional changes. Moreover, the use of collaborative training to optimize the distribution prediction module enables accurate prediction of future distributional changes.

3. Experimental results across various datasets and benchmark models demonstrate the efficacy of the DDN, showing significant improvement over traditional normalization methods. This thoroughly validates the applicability and superiority of the proposed method.

4. The paper is well-organized. The tables, figures and notations are very clear.

**Weaknesses:**

1. The paper suggests that using proper windows for low and high-frequency components, which can better capture distributional changes. However, this lacks comparative experiments. A comparison with the traditional approach of using a uniform large window for both high and low-frequency components would better demonstrate the superior ability of the proposed method in capturing dynamic changes.
2. The terminology introduced should be consistent with previous works for easier readability. While the concept of sliding statistics is recommended to replace with a more unified term "rolling statistic", as employed in the referenced paper [1].
3. Some symbols need correction. In Section 3.3, the mean symbols (μ_f^i、σ_f^i) of the mean and standard deviation sequences in the Frequency Domain Prediction part are inconsistent with Figure 3. Additionally, the symbols in Figure 3 appear more like the Time Domain Prediction.
4. The proposed method's comparison is primarily focused on the current four mainstream models. It is recommended to include more baseline models for comparison, such as Transformer and PatchTST, in the experimental section.

[1] Eric Zivot, Jiahui Wang, Eric Zivot, and Jiahui Wang. Rolling analysis of time series. Modeling financial time series with S-Plus®, pages 299–346, 2003. 2

**Questions:**

1. What is effects of window size for low and high-frequency components?
2. Does the work also work for other SOTA methods, like PatchTST?

**Limitations:**

Please refer to the weaknesses above.

---

> ### Author Rebuttal · Authors · 2024-08-07
>
> Thanks for your valuable comments on our work, we will carefully review and revise our manuscript to correct any inappropriate expressions or symbol errors. Here are responses to your concerns and questions:
>
> **Question 1:** What is effects of window size for low and high-frequency components?
>
> **Response 1:**
> To address your concerns, we evaluated the impact of sliding window sizes on model performance using multiple window sizes. Similar to existing work [1], a large sliding window for the low-frequency component captures long-term changes with minimal bias and eliminates local high-frequency fluctuations, while a small sliding window for the high-frequency component targets short-term rapid changes to capture rapid variations.
> |L=96|iTransformer||||||||
> |-|-|-|-|-|-|-|-|-|
> |Size|(7，7)|(7，7)|(7，12)|(7，12)|(12，12)|(12，12)|(7，24)|(7，24)|
> |Metric|MSE|MAE|MSE|MAE|MSE|MAE|MSE|MAE|
> |Electricity|||||||||
> |96|0.133|0.233|0.131|0.231|0.132|0.231|0.127|0.225|
> |192|0.152|0.253|0.149|0.249|0.149|0.25|0.146|0.246|
> |336|0.157|0.261|0.156|0.258|0.157|0.26|0.156|0.257|
> |720|0.187|0.291|0.180|0.282|0.182|0.285|0.179|0.282|
> |Traffic|||||||||
> |96|0.342|0.252|0.336|0.248|0.338|0.249|0.341|0.252|
> |192|0.353|0.258|0.347|0.254|0.348|0.256|0.348|0.257|
> |336|0.374|0.268|0.363|0.263|0.365|0.264|0.367|0.265|
> |720|0.433|0.296|0.412|0.286|0.438|0.306|0.418|0.296|
>
> Here, (7, 12) indicates a window size of 7 for the high-frequency component, and the remaining window size of 12. Additionally, we will supplement our study with more relevant experiments to further alleviate the your concerns.
>
> **Question 2:** Does the work also work for other SOTA methods, like PatchTST?
>
> **Response 2:**
> We will supplement our study with more mainstream baselines currently in use. Below are additional baselines commonly used for reversible normalization. Additionally, we will include more comprehensive baselines for PatchTST, Crossformer, and SCNN, as these are of particular interest to the reviewers. *OOM* means out of memory.
> ||||||||||||||
> |-|-|-|-|-|-|-|-|-|-|-|-|-|
> |L=96|PatchTST|PatchTST|+DDN|+DDN|Transformer|Transformer|+DDN|+DDN|Informer|Informer|+DDN|+DDN|
> |Metric|MSE|MAE|MSE|MAE|MSE|MAE|MSE|MAE|MSE|MAE|MSE|MAE|
> |Weather|||||||||||||
> |96|0.147|0.197|0.147|0.199|0.37|0.435|0.188|0.235|0.424|0.455|0.187|0.237|
> |192|0.191|0.240|0.190|0.239|0.513|0.490|0.238|0.286|0.421|0.444|0.246|0.287|
> |336|0.244|0.282|0.241|0.283|0.702|0.590|0.314|0.343|0.579|0.536|0.309|0.335|
> |720|0.320|0.334|0.305|0.330|0.853|0.691|0.401|0.397|0.945|0.729|0.390|0.391|
> |Electricity|||||||||||||
> |96|0.138|0.233|0.133|0.231|0.258|0.359|0.165|0.269|0.316|0.403|0.188|0.295|
> |192|0.153|0.247|0.147|0.245|0.262|0.360|0.185|0.285|0.354|0.434|0.213|0.318|
> |336|0.170|0.263|0.164|0.262|0.285|0.380|0.200|0.301|0.372|0.447|0.221|0.326|
> |720|0.206|0.296|0.195|0.293|0.288|0.374|0.220|0.322|0.392|0.451|0.250|0.351|
> |Traffic|||||||||||||
> |96|OOM|OOM|OOM|OOM|0.684|0.381|0.515|0.312|0.725|0.414|0.566|0.359|
> |192|OOM|OOM|OOM|OOM|0.659|0.360|0.528|0.319|0.748|0.423|0.589|0.373|
> |336|OOM|OOM|OOM|OOM|0.653|0.352|0.545|0.332|0.865|0.498|0.634|0.405|
> |720|OOM|OOM|OOM|OOM|0.675|0.365|0.584|0.345|1.004|0.556|0.676|0.418|
>
> [1] J. Deng et al., "Disentangling Structured Components: Towards Adaptive, Interpretable and Scalable Time Series Forecasting," in IEEE Transactions on Knowledge and Data Engineering, doi: 10.1109/TKDE.2024.3371931.

---

> > ### Comment · Reviewer_qBzF · 2024-08-09
> > **Thanks for the response!**
> >
> > Thanks for the detailed response. My concerns are well addressed. I acknowledge the novelty of using the inherent differences in frequency and the sliding statistics in dynamically capturing distributional changes over time.

---

> > > ### Author Response · Authors · 2024-08-11
> > > **Reply to reviewer**
> > >
> > > Thanks a lot for your valuable comments and recognition for our work.

---

### Official Review · Reviewer_Ej1X · 2024-07-10

**Soundness:** 4
**Presentation:** 3
**Contribution:** 3
**Rating:** 7
**Confidence:** 4

**Summary:**

The authors consider the data distribution variations for real-world data and then propose a novel dual-domain dynamic normalization. Unlike the previous methods work in time domain, the proposed method decompose time series into a linear combination of different frequencies, and dynamically capture distribution variations in both time and frequency domains. Besides, the proposed method can serve as a plug-in-play module, and thus can be easily incorporated into other forecasting models. Extensive experiments on public benchmark datasets under different forecasting models demonstrate the effectiveness of the proposed method.

**Strengths:**

1. The motivation is reasonable by eliminating the non-stationarity of time series via both frequency and time domain normalization in a sliding window way. The way of dual-domain extraction seems effective, compared with the previous methods in individual time domain.
2. The proposed method works well in eliminating non-stationary factors with frequency domain normalization and time domain normalization. Benefiting from the complementary properties of the time and frequency domain information, it allows the proposed method to further clarify non-stationary factors and reconstruct non-stationary information.
3. Extensive experiments demonstrate the effectiveness of the proposed method, by achieving significant performance improvements across various baseline models on seven real-world datasets.
4. The presentation is well-written and easy to follow.

**Weaknesses:**

1. The proposed method tries to decompose the original time series by low and high-frequency components. However, how to decompose time series has less explored.
2. The proposed Dual-domain method mainly considers time series in both frequency and time domain normalization. However, the effect of domains has less explore. For example, we can also transform the time series into high-dimension latent space.
3. Compared with other normalizaiton methods, the advantages and disadvantages should be further discussed.

**Questions:**

Please also see the Weaknesses.
1. What is effects of the way of decomposition methods?
2. What is effects of different domains, such as time/frequency/embedding space?
3. What are the advantages and disadvantages of the proposed methods.

**Limitations:**

The limitations have been discussed and there are no potential negative ethical and societal implications in this work.

---

> ### Author Rebuttal · Authors · 2024-08-07
>
> Thanks for your valuable comments of our work. Here are responses to your concerns and questions:
>
> **Question 1:** What is effects of the way of decomposition methods?
>
> **Response 1:** As highlighted by the work [1] of reviewer G9hB, for long-term distribution changes, a larger sliding window is required to eliminate local high-frequency fluctuations and obtain an estimate of the long-term distribution characteristics with minimal bias. Conversely, for short-term distribution changes, a smaller sliding window is necessary to fully capture the complex dynamics. Our approach to decomposition is similar, utilizing Wavelet Transform to separate high-frequency and low-frequency components. The low-frequency component naturally separates out local high-frequency fluctuations, allowing us to use a larger sliding window to estimate long-term distribution characteristics. Meanwhile, the high-frequency component inherently contains rapid dynamic changes, enabling us to use a smaller sliding window to quickly capture short-term distribution variations.
>
> **Question 2:** What is effects of different domains, such as time/frequency/embedding space?
>
> **Response 2:**
> In our study, we combine time and frequency domains. The frequency domain is used to enhance the extraction of short-term high-frequency non-stationary features, while the time domain is utilized to enhance the diversity of distribution characteristics during DPM training, simultaneously avoiding interference from high-frequency noise in the extracted distribution features.
> Regarding the embedding space, although previous work [2] has demonstrated that linear projection can extract coarse-grained distribution characteristics from the latent space and theoretically avoid interference from the adopted frequencies while achieving higher accuracy, we did not adopt this method. The reason lies in the complex variations of distribution characteristics, which make it difficult for a simple projection operation to accurately capture fine-grained distribution features.
>
> **Question 3:** What are the advantages and disadvantages of the proposed methods.
>
> **Response 3:**
> Our method has several advantages:
>
> 1) It can be directly integrated with all current models.
>
> 2) It effectively addresses the challenges posed by non-stationary data in time series prediction.
>
> 3) It maintains SOTA performance compared to a range of current reversible normalization methods.
>
> However, our method also has some disadvantages:
>
> 1) DPM is an MLP-based network, which adds extra parameters.
>
> 2) Precisely capturing fine-grained distribution characteristics in the embedding space using a data-driven approach remains an unresolved issue.
>
> [1] J. Deng et al., "Disentangling Structured Components: Towards Adaptive, Interpretable and Scalable Time Series Forecasting," in IEEE Transactions on Knowledge and Data Engineering, doi: 10.1109/TKDE.2024.3371931.
>
> [2] Fan W, Wang P, Wang D, et al. Dish-ts: a general paradigm for alleviating distribution shift in time series forecasting[C]//Proceedings of the AAAI conference on artificial intelligence. 2023, 37(6): 7522-7529.

---

> > ### Comment · Reviewer_Ej1X · 2024-08-12
> > **Good response**
> >
> > Thanks for your response. My concerns have been addressed. I'd like to keep my score to 7 (accept).

---

### Official Review · Reviewer_i5Qt · 2024-07-12

**Soundness:** 3
**Presentation:** 2
**Contribution:** 2
**Rating:** 5
**Confidence:** 3

**Summary:**

The paper introduces a approach to improve the accuracy of time series forecasting by addressing the challenge of non-stationary data, where data distributions change rapidly over time. The authors propose a Dual-domain Dynamic Normalization (DDN) framework that captures distribution variations dynamically in both time and frequency domains using Discrete Wavelet Transform (DWT) and sliding normalization. This method enhances the robustness of forecasting models against non-stationary data, significantly outperforming existing normalization methods in extensive experiments on public benchmark datasets.

**Strengths:**

1. The paper proposes a highly effective method for handling non-stationary time series data by dynamically capturing distribution variations in both time and frequency domains, addressing a significant challenge in time series forecasting.
2. The quality of the experimental results is strong, with extensive tests on seven public benchmark datasets showing that DDN consistently outperforms existing normalization methods across various forecasting models, demonstrating significant improvements in prediction accuracy.
3. The method's practical utility is notable as it can be easily integrated into existing forecasting models, making it a versatile tool for improving the reliability and accuracy of time series forecasts in diverse real-world applications.

**Weaknesses:**

1. **Lack of Hyperparameter Sensitivity Analysis**: The paper does not include a sensitivity analysis of the hyperparameters, such as the length of the sliding window.
2. **Insufficient Theoretical and Empirical Justification**: The paper primarily focuses on empirical results without providing a strong theoretical foundation for the proposed Dual-domain Dynamic Normalization (DDN) framework. Additionally, there is a lack of ablation studies or detailed experiments that demonstrate the specific contributions and effectiveness of each module within the DDN framework. This makes it challenging to understand the individual impact of each component and the underlying principles that contribute to the overall performance improvements.
3. **Inconsistent Look-back Window Selection**: The paper does not explain why different models use different look-back window sizes or how these variations impact the experimental results. The lack of rationale behind the choice of look-back windows and the absence of an analysis of their effects on model performance makes it difficult to assess the consistency and fairness of the comparisons.

**Questions:**

1. Could you provide a sensitivity analysis of the hyperparameters, such as the length and stride of the sliding window? Understanding how these parameters affect performance is crucial.
2. Could you share experimental results that illustrate the impact of different look-back window sizes on model performance? This would clarify how the choice of look-back windows influences the results.
3. Could you offer theoretical or empirical analyses to demonstrate the contributions and effectiveness of each module within the DDN framework? Detailed ablation studies or theoretical justifications would be very helpful.

---

> ### Author Rebuttal · Authors · 2024-08-07
>
> Thanks for your valuable comments of our work. Here are responses to your concerns and questions:
>
> **Question 1:** Lack of Hyperparameter Sensitivity Analysis, such as the length of the sliding window.
>
> **Response 1:** As suggested, we conduct experiments to evaluate the impact of sliding window size of our model. In the table below, we compare results with various sliding window sizes. For example, (7,12) represents sliding window sizes of 7 and 12 for high and low-frequency components.
> ||iTransformer||||||||
> |-|-|-|-|-|-|-|-|-|
> |Size|(7，7)|(7，7)|(7，12)|(7，12)|(12，12)|(12，12)|(7，24)|(7，24)|
> |Metric|MSE|MAE|MSE|MAE|MSE|MAE|MSE|MAE|
> |Electricity|||||||||
> |96|0.133|0.233|0.131|0.231|0.132|0.231|0.127|0.225|
> |192|0.152|0.253|0.149|0.249|0.149|0.25|0.146|0.246|
> |336|0.157|0.261|0.156|0.258|0.157|0.26|0.156|0.257|
> |720|0.187|0.291|0.180|0.282|0.182|0.285|0.179|0.282|
> |Traffic|||||||||
> |96|0.342|0.252|0.336|0.248|0.338|0.249|0.341|0.252|
> |192|0.353|0.258|0.347|0.254|0.348|0.256|0.348|0.257|
> |336|0.374|0.268|0.363|0.263|0.365|0.264|0.367|0.265|
> |720|0.433|0.296|0.412|0.286|0.438|0.306|0.418|0.296|
>
> As table we can see that a large sliding window for low-frequency components and a small one for high-frequency components often yields better results, which aligns with the viewpoint of existing work [1] mentioned by Reviewer G9hB.
>
> Besides, we perform sensitivity analysis on other hyperparameters, including different strides and initial wavelet bases. It is observed that our performance improves when the stride is close to a point level, and different initial wavelet bases do not significantly affect the overall results. We would add these analysis in revision.
> |L=336|DLinear||||||
> |-|-|-|-|-|-|-|
> |stride|1|1|4|4|7|7|
> |Metric|MSE|MAE|MSE|MAE|MSE|MAE|
> |Weather|0.223|0.274|0.225|0.275|0.226|0.275|
> |Electricity|0.165|0.263|0.167|0.265|0.168|0.268|
> |Traffic|0.421|0.291|0.425|0.294|0.427|0.296|
>
> |L=336|DLinear||||||
> |-|-|-|-|-|-|-|
> |Wavelet|coiflet3|coiflet3|sym3|sym3|db3|db3|
> |Metric|MSE|MAE|MSE|MAE|MSE|MAE|
> |Weather|0.223|0.274|0.222|0.273|0.225|0.277|
> |Electricity|0.165|0.263|0.165|0.262|0.167|0.266|
> |Traffic|0.421|0.291|0.421|0.292|0.424|0.294|
>
> **Question 2:** Insufficient Theoretical and Empirical Justification.
>
> **Response 2:**
> Our work is based on widely adopted theoretical principle, validated by thorough empirical evaluation：
>
> 1) On the theoretical side, our method draws inspiration from rolling statistics and frequency domain analysis. Particularly, rolling statistics serve as the theoretical foundation of many time series analysis and normalization works [1, 2].
>
> 2) On the empirical side, the importance and effectiveness of each component are demonstrated by the ablation study. Specifically, in Section 4.3 we experiment with using only the frequency or the time domain branch (for convenience, the table is presented below). The results show that the combination of these two branches typically outperforms a single branch, and the frequency domain branch usually outperforms the time domain branch.
> ||||DLinear||||||iTransformer||||
> |-|-|-|-|-|-|-|-|-|-|-|-|-|
> ||DDN|DDN|Frequency|Frequency|Time|Time|DDN|DDN|Frequency|Frequency|Time|Time|
> |Metric|MSE|MAE|MSE|MAE|MSE|MAE|MSE|MAE|MSE|MAE|MSE|MAE|
> |Weather|0.227|0.273|0.224|0.273|0.231|0.283|0.225|0.280|0.221|0.270|0.219|0.272|
> |Electricity|0.162|0.260|0.161|0.259|0.155|0.255|0.152|0.252|0.161|0.259|0.152|0.252|
> |Traffic|0.414|0.284|0.409|0.280|0.368|0.265|0.366|0.262|0.408|0.278|0.364|0.263|
>
> 3) Each component of our method is designed to address specific challenges. First, the frequency branch addresses the ignorance of high-frequency changes. Second, the time branch aims at providing multi-scale distribution statistics and balancing the high-frequency noise that may amplified in the frequency branch. Finally, these two branches are combined in a unified framework to maximize their complementary advantages.
>
> **Question 3:** Inconsistent Look-back Window Selection
>
> **Response 3:**
> Thanks for pointing it out. **Inconsistent look-back window sizes do not affect the fairness of our comparison**, due to the following reasons:
>
> 1. Our work presents a plugin normalization module, and the focus is on performance improvement after its use. Therefore, maintaining the same look-back window size is sufficient. Similar practices can be found in existing works [3,4].
>
> 2. Different baselines have their best performance with different look-back window sizes, so using a unified window size cannot demonstrate their full potential. For example, CI methods tend to use larger look-back windows (e.g., DLinear and PatchTST use a 336 look-back window, while Autoformer and Fedformer use 96).
>
> For an intuitive comparision, we also conduct  experiments using the same look-back window size of 96. The results are shown in the table below, from which we see that our method still obtains better results. These further demonstrate the superiority of our method.
>
> |L=96|DLinear|DLinear|DDN|DDN|iTransformer|iTransformer|DDN|DDN|
> |-|-|-|-|-|-|-|-|-|
> |Metric|MSE|MAE|MSE|MAE|MSE|MAE|MSE|MAE|
> |Electricity|||||||||
> |96|0.197|0.282|0.167|0.259|0.148|0.24|0.14|0.238|
> |192|0.196|0.285|0.178|0.271|0.162|0.253|0.157|0.25|
> |336|0.209|0.301|0.193|0.288|0.178|0.269|0.173|0.267|
> |720|0.245|0.333|0.228|0.32|0.225|0.317|0.208|0.31|
> |Traffic|||||||||
> |96|0.65|0.396|0.476|0.297|0.395|0.268|0.392|0.265|
> |192|0.598|0.37|0.488|0.299|0.417|0.276|0.413|0.273|
> |336|0.604|0.373|0.507|0.306|0.433|0.283|0.424|0.278|
> |720|0.645|0.394|0.55|0.326|0.467|0.302|0.444|0.29|
>
> [1] J. Deng et al. Disentangling Structured Components: Towards Adaptive, Interpretable and Scalable Time Series Forecasting.
>
> [2] Zivot E et al. Rolling analysis of time series
>
> [3] Liu Z, et al. Adaptive normalization for non-stationary time series forecasting: A temporal slice perspective.
>
> [4] Han L, et al. SIN: Selective and Interpretable Normalization for Long-Term Time Series Forecasting.

---

> ### Author Response · Authors · 2024-08-11
> **Reminder for post-rebuttal feedback**
>
> Dear Reviewer i5Qt
>
> We greatly appreciate your initial valuable comments. We hope that you could have a quick look at our responses to your concerns. It would be highly appreciated if you could kindly update the initial rating if your questions have been addressed. We are also happy to answer any additional questions before the rebuttal ends.
>
> Best regards

---

> ### Comment · Reviewer_i5Qt · 2024-08-11
>
> Thank you for your response, my concerns have been resolved, and considering other reviewers' concerns as well, I have raised my score to 5. Also, it's better to include some of the analysis you provided above in the paper for readers to understand the method further.

---

> > ### Author Response · Authors · 2024-08-14
> >
> > Thank you for your valuable comments and  positive feedback for acknowledging our efforts in addressing your concerns. As suggested, we would add more experiments and analysis we provided in our revised manuscript.

---

### Decision · Program_Chairs · 2024-09-25

**Decision:**

Accept (poster)

**Comment:**

The paper introduces a novel Dual-domain Dynamic Normalization (DDN) framework to address the challenges of non-stationary time series forecasting. By dynamically capturing distribution variations in both time and frequency domains, DDN can enhance the robustness and accuracy of forecasting models. The proposed method is validated through extensive experiments on benchmark datasets, outperforming existing state-of-the-art normalization techniques.

During the rebuttal period, reviewers raised concerns regarding hyperparameter analysis, effects of the way of decomposition methods, more comparisons with SOTA methods, look-back window selection, etc. The authors provided thorough responses with extra experiments, which satisfactorily addressed most of these concerns. While some minor weaknesses exist, they are not critical and can be addressed in the CR version. Given the experimental results and the overall positive feedback from the reviewers (with scores of 7, 7, 5, 5), I recommend accepting the paper.